# Debiased, Longitudinal and Coordinated Drug Recommendation through Multi-Visit Clinic Records

**Hongda Sun, Shufang Xie, Shuqi Li, Yuhan Chen, Ji-Rong Wen, Rui Yan**[*]
Gaoling School of Artificial Intelligence, Renmin University of China, Beijing, China
{sunhongda98,shufangxie,shuqili,yuhanchen,jrwen,ruiyan}@ruc.edu.cn

## Abstract

AI-empowered drug recommendation has become an important task in healthcare research areas, which offers an additional perspective to assist human doctors with more accurate and more efficient drug prescriptions. Generally, drug recommendation is based on patients' diagnosis results in the electronic health records. We assume that there are three key factors to be addressed in drug recommendation: 1) elimination of recommendation bias due to limitations of observable information, 2) better utilization of historical health condition and 3) coordination of multiple drugs to control safety. To this end, we propose DrugRec, a causal inference based drug recommendation model. The causal graphical model can identify and deconfound the recommendation bias with front-door adjustment. Meanwhile, we model the multi-visit in the causal graph to characterize a patient's historical health conditions. Finally, we model the drug-drug interactions (DDIs) as the propositional satisfiability (SAT) problem, and solving the SAT problem can help better coordinate the recommendation. Comprehensive experiment results show that our proposed model achieves state-of-the-art performance on the widely used datasets MIMIC-III and MIMIC-IV, demonstrating the effectiveness and safety of our method.

## 1 Introduction

The digitization of healthcare track records with diagnosis, procedure, and prescriptions has become an inevitable trend rising. Advanced AI can process large-scale electronic health records (EHRs) and biomedical knowledge graphs (BioKGs) with the massive data available. With the continuous progress of medical digitization, the advances in deep learning technologies for processing can assist doctors and researchers in making accurate and efficient medical decisions [26, 18]. More specifically, the goal of the drug recommendation task is to recommend appropriate medication combinations based on a patient's disease conditions. Existing drug recommendation methods model a patient based on his/her diagnoses and procedures in health records from actual hospital visits and then determine which drugs to recommend [27, 25]. The information on BioKGs is also widely used in the drug recommendation process to avoid the negative effect of drug-drug interactions (DDIs) [17, 28].

Despite the promising achievements of previous methods, we argue that three essential factors should be considered in a drug recommendation model. First, general recommendation systems often suffer from recommendation bias issues [1] (e.g., conformity bias, exposure bias, popularity bias, selection bias, etc.), which also happens in drug recommendations but is not handled well in previous works. Sometimes the diagnostic information in EHRs is insufficient to describe the patient's health condition, leading to a recommendation bias due to the limitations of observable information. Second, the patient's historical clinic records are critical for recommending drugs effectively. When recommending drugs, the model should consider both the current visit and previous

---

[*]Corresponding author: Rui Yan (ruiyan@ruc.edu.cn)

36th Conference on Neural Information Processing Systems (NeurIPS 2022).

visits, since ignoring them could lead to a lower recommendation accuracy. Finally, the negative drug-drug interactions (DDI) is also a critical issue in drug recommendation models. When multiple drugs are recommended together, the system needs to control DDIs for drug safety.

Considering these key factors, we propose a novel drug recommendation causal graphical model, which is shown in Figure 1. The patient's potential disease information is considered as the *confounder* ($C$). Due to the existence of the confounder, general drug recommendation leads to recommendation bias due to ignoring the existence of the confounder. We debias the confounding defects by identifying the causal effect of the *treatment* variable symptom ($S$) on the *outcome* variable ($Y$). Identifying the effect can take into account the existence of the confounder in modeling to eliminate the recommendation bias. Specifically, we intervene on the symptom and identify the deconfounded causal effect of symptom on drug prescriptions with front-door adjustment [5]. As a result, we can obtain debiased drug recommendation results by maximizing the average treatment effect (ATE) of intervention. Moreover, in order to extend the method to the multi-visit scenario, we propose two modeling schemes to leverage the historical records for current drug prescribing. Therefore, we can successfully identify the multi-visit causal effect for more accurate recommendation. Finally, we model the DDI problem as the propositional satisfiability (SAT) problem where the selection of each drug is represented as a boolean variable and the relations are represented as boolean operators. Although the SAT is NP-complete [3, 12], we note the DDI problem can be modelled as a particular case, i.e., 2-SAT problem, which can be solved in polynomial time [10]. Since the trivial solution (i.e., recommend no drugs) always exists, we propose a heuristic method to extract non-trivial 2-SAT solution that optimize the recommendation probability from the causal graph.

Our overall technical contributions in this work are summarized as follows:

- We construct a drug recommendation causal graphical model for this task and leverage the front-door adjustment to alleviate the invisible recommendation bias.
- We propose two modeling schemes that extend the graphical model to the multi-visit scenario to better model a patient's historical health condition.
- We model the DDI with the 2-SAT problem to coordinate the recommendation and improve the recommendation safety. We also propose a heuristic method to extract non-trival solution to 2-SAT to optimize the recommendation probability from causal graph.

We conduct extensive experiments on the widely used benchmark datasets MIMIC-III and MIMIC-IV to evaluate the recommendation quality and the DDI rate. We significantly (p<0.01) advance the state-of-the-art results of Jaccard score, PRAUC, and F1-score, demonstrating the effectiveness of our method. The significant drop in DDI Rate shows that our method better considers drug safety.

## 2 Related Work

**Drug Recommendation.** Drug recommendation is a promising research area in recent years. For example, Gong et al. [6] construct a high-quality heterogeneous graph and decomposes the medicine recommendation into a link prediction process. Zhang et al. [30] decompose treatment recommendation task into a sequential decision-making process and adopt a multi-instance multi-label learning framework. Shang et al. [16] pre-train the records of single-visit patient's records and fine-tune on the multi-visit records. Choi et al. [2] employ a two-level neural attention model that detects influential past visits and significant clinical variables. Le et al. [11] present a new memory augmented neural network model to model sequential records. Shang et al. [17] utilize memory augmented neural networks and store historical records as references for future prediction. Yang et al. [27] first focus on the medication changes using recurrent residual learning . Yang et al. [28] propose a DDI-controllable drug recommendation model to leverage molecule structures and model DDIs more effectively. Wu et al. [25] introduces a novel copy-or-predict mechanism to generate medicines. We conduct a novel causal graoh for drug recommendation treating the recommendation bias as a confounder, and make debiased recommendations by identifying the causal effect.

**Causal Recommendation.** Causal inference has been widely used in machine learning based recommendation systems. The confounding effect is the typical pattern for considering causality in recommendation systems. Wang et al. [22] employ deconfounding techniques to learn real interests affected by unobserved confounders. The recommender estimates a substitute for the unobserved confounders by fitting exposure data. Sato et al. [15] treat the features of the user

and item as confounders and reweight training samples to account for confounding. Many existing methods construct causal graphs and then apply causal inference techniques to mitigate confounding bias by incorporating confounders into data generation, including item popularity [29, 23, 7], user selection [21, 22], and ranking positions [13]. We are the first to design a novel causal graph for drug recommendation from the logic of drug prescribing of human doctors, which eliminates the recommendation bias due to limitations of observable information.

# 3 Preliminary

## 3.1 Problem Formulation

Longitudinal EHR data has the patient-visit hierarchical structure and the whole data for all patients can be represented as $\mathbf{X} = \{\mathbf{X}^{(1)}, \mathbf{X}^{(2)}, \ldots, \mathbf{X}^{(N)}\}$, where $N$ is the total number of patients. A specific patient $j$ can be represented as a sequence of multivariate visits: $\mathbf{X}^{(j)} = [\mathbf{x}_1^{(j)}, \mathbf{x}_2^{(j)}, \ldots, \mathbf{x}_{T_j}^{(j)}]$, where $T_j$ is the number of visits of patient $j$. For the $t$-th visit of patient $j$, $\mathbf{x}_t^{(j)} = [\mathbf{s}_t^{(j)}, \mathbf{d}_t^{(j)}, \mathbf{p}_t^{(j)}, \mathbf{m}_t^{(j)}]$, where $\mathbf{s}_t^{(j)} \in \{0,1\}^{|\mathcal{S}|}$, $\mathbf{d}_t^{(j)} \in \{0,1\}^{|\mathcal{D}|}$, $\mathbf{p}_t^{(j)} \in \{0,1\}^{|\mathcal{P}|}$ and $\mathbf{m}_t^{(j)} \in \{0,1\}^{|\mathcal{M}|}$ are multi-hot symptoms, diagnoses, procedures, and medication vectors, respectively. The $\mathcal{S}$, $\mathcal{D}$, $\mathcal{P}$ and $\mathcal{M}$ are the overall symptom, diagnosis, procedure, and medication sets, while $|\cdot|$ denotes the cardinality of the set. We will omit the trivial superscript $j$ whenever there is no ambiguity in discussing different visits within the same patient. Meanwhile, we also need to consider the side effect relations in the DDI graph before determining the final drug combinations. We use the symmetric binary adjacency matrix $\mathbf{A} \in \{0,1\}^{|\mathcal{M}|\times|\mathcal{M}|}$ to represent the DDI graph, where $\mathbf{A}_{uv} = 1$ if and only if the $u$-th and $v$-th drug has harmful interactions.

The objective of drug recommendation task is to predict the probabilities $\widehat{\mathbf{y}}_t \in [0,1]^{|\mathcal{M}|}$ of each drug to recommend, given all the patient's previous visits $(\mathbf{x}_1, \mathbf{x}_2, \ldots, \mathbf{x}_{t-1})$, current visit inputs $(\mathbf{s}_t, \mathbf{d}_t, \mathbf{p}_t)$ at time $t$, and the DDI graph $\mathbf{A}$.

## 3.2 Causal View of Drug Recommendation

To address the problem that observed information is often incomplete and insufficient to describe the actual health condition, we propose a debiased drug recommendation method based on causal inference techniques. As shown in Figure 1, it is consistent with the logic of doctors diagnosing diseases based on symptoms and prescribing drugs based on all the information. we introduce an unobservable confounder as the patient's potential disease information, which acts directly on both symptoms $(S)$ and drug prescriptions $(Y)$, affecting the results of drug recommendation. In the causal view, the conditional probability $P(y|s, m)$ considered by general models cannot reflect the actual causal effect of symptoms on drug prescribing. We intervene on the treatment variable $S$, and the intervention probability $P(y|do(s), m)$ determines the effect of $S$ on $Y$. Next, the key to identifying the causal effect is how to remove the *do*-operator so that $P(y|do(s), m)$ can be calcu-

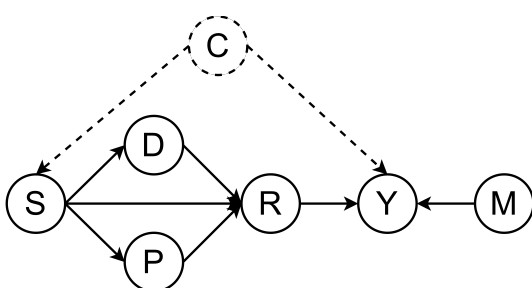

Figure 1: The causal graph of drug recommendation. The variables represent: C: confounder, D: diagnosis, P: procedure, R: patient visit representation, S: symptom, M: medication, Y: recommendation probability. The dotted arrows and circles represent latent variable and links.

lated. Here we treat $D$, $P$ and $R$ as *mediators*, which exactly satisfies the conditions of the front-door criterion [5]. The front-door criterion provides a sufficient condition for identifiability when the confounder is unobservable. The causal effect $P(y|do(s), m)$ is thus identifiable with front-door adjustment [5], which can be formulated as Equation (1). See Appendix A for details of causal analysis.

$$
\begin{aligned}
P(y|do(s), m) &= \sum_{r \in \mathcal{R}} \sum_{d \in \mathcal{D}} \sum_{p \in \mathcal{P}} P(d|s) P(p|s) P(r|s, d, p) \sum_{s' \in \mathcal{S}} P(y|s', r, m) P(s') \\
&\triangleq \sum_{s' \in \mathcal{S}} f(s', r(s, d_s, p_s), m) P(s').
\end{aligned}
\tag{1}
$$

Denote $d_s$ and $p_s$ as symptom-affected representations of the diagnosis and procedure, respectively. We set $P(d|s) = 1$ if and only if $d = d_s$; otherwise $P(d|s) = 0$, and $P(p|s)$ as the same. The patient visit representation is written as $r(s, d_s, p_s)$ under the representations $s, d_s, p_s$, and $P(r|s, d_s, p_s) = 1$ holds if and only if $r = r(s, d_s, p_s)$; otherwise $P(r|s, d_s, p_s) = 0$. The objective of previous methods that ignore modeling the unobserved confounder is to learn a scoring function of the form $f(r(d, p), m)$ or $f(r(s, d_s, p_s), m)$. In contrast, our scoring function $f$ is related to the representation of each symptom $s'$. The eventual probability is a weighted sum of the scoring functions according to different symptoms. We can finally estimate $\widehat{\mathbf{y}} \in [0, 1]^{|\mathcal{M}|}$, the predicted recommendation probability for each drug, by Equation (1). Since the cardinality of the symptom set $|\mathcal{S}|$ is large, it is untrackable to traverse all the symptoms to calculate the above probability. Instead, we conduct a sample set $\hat{S}$ with $k_s$ symptoms. The probability $P(s')$ can be estimated by the frequency $\hat{P}(s')$ for symptom $s'$ in the original EHRs.

## 4 Our method

### 4.1 Multi-Visit Causal Drug Recommendation

The causal graph in Figure 1 is based on a single visit of a patient. Considering the critical impact of a patient's historical health condition on current drug recommendations, we extend the single-visit causal effect to the multi-visit scenario. We propose two modeling schemes for the multi-visit causal drug recommendation. The **DrugRec-a** aggregates *all* the historical visits and express them as one causal graph, while the **DrugRec-k** models the impact of past $k$ visits before the current visit.

**DrugRec-a.** A core issue of the previous single-visit scenario is that the causal effects of each visit are calculated separately without any interaction between them. Considering that variables like $S$, $D$, $P$ and $Y$ have sequential effects, the DrugRec-a method integrates all previous variable representations of $t - 1$ visits into a single representation at time $t$ using an aggregation function $agg$. Therefore, it can be uniformly handled that the lengths of historical visits vary at different time points. For instance, we denote the integrated symptom representation as $\overline{S}_{t-1} = agg([S_1, \ldots, S_{t-1}])$, and other integrated representations $\overline{D}_{t-1}, \overline{P}_{t-1}, \overline{Y}_{t-1}$ can be derived in the same way. Therefore, we can compress all previous $t - 1$ visits into one causal graph. Every time the model makes decisions for the current drug prescription, the connection between the historical causal graph and the current causal graph is crucial. We link the two causal graphs together by adding several paths between them. Specifically, we link the paths $\overline{S}_{t-1} \rightarrow S_t$, $\overline{D}_{t-1} \rightarrow D_t$, $\overline{P}_{t-1} \rightarrow P_t$ and $\overline{Y}_{t-1} \rightarrow Y_t$ for later joint analysis. Similar to the single-visit scenario, the multi-visit causal effect is now extended to a joint intervention probability $P(y_t, \overline{y}_{t-1}|do(s_t), do(\overline{s}_{t-1}), m)$ for the historical and current patient health conditions. Historical and current diagnoses, procedures and patient visit representations can still serve as mediators, as all pathways from treatment variables $(\overline{S}_{t-1}, S_t)$ to outcome variables $(\overline{Y}_{t-1}, Y_t)$ pass through them. Thus, the overall linked causal graph still satisfies the conditions of the front-door criterion. More specifically, the causal effect $P(y_t, \overline{y}_{t-1}|do(s_t), do(\overline{s}_{t-1}), m)$ can be identifiable and formulated as

$$P(y_t, \overline{y}_{t-1}|do(s_t), do(\overline{s}_{t-1}), m) = \sum_{\overline{s}'_{t-1} \in \mathcal{S}} f\left(\overline{s}'_{t-1}, r\left(\overline{s}_{t-1}, \overline{d}_{t-1,\overline{s}_{t-1}}, \overline{p}_{t-1,\overline{s}_{t-1}}\right), m\right) P(\overline{s}'_{t-1})$$
$$\cdot \sum_{r_t \in \mathcal{R}} \sum_{d_t \in \mathcal{D}} \sum_{p_t \in \mathcal{P}} P\left(d_t|\overline{d}_{t-1}, s_t\right) P(p_t|\overline{p}_{t-1}, s_t) P(r_t|s_t, d_t, p_t)$$
$$\cdot \sum_{s'_t \in \mathcal{S}} P(y_t|\overline{y}_{t-1}, s'_t, r_t, m) P(s'_t|\overline{s}'_{t-1}). \tag{2}$$

We rearrange the Equation (2) and decompose it into the following two parts, corresponding to the patient's historical and current visit.

$$P(y_t, \overline{y}_{t-1}|do(s_t), do(\overline{s}_{t-1}), m) \triangleq \sum_{\overline{s}'_{t-1} \in \mathcal{S}} f\left(\overline{s}'_{t-1}, r\left(\overline{s}_{t-1}, \overline{d}_{t-1,\overline{s}_{t-1}}, \overline{p}_{t-1,\overline{s}_{t-1}}\right), m\right) P(\overline{s}'_{t-1})$$
$$\cdot \sum_{s'_t \in \mathcal{S}} \widetilde{f}\left(\overline{y}_{t-1}, s'_t, r\left(\widetilde{s}_t, \widetilde{d}_{t,s_t}, \widetilde{p}_{t,s_t}\right), m\right) P(s'_t|\overline{s}'_{t-1}), \tag{3}$$

where $P(d_t|\overline{d}_{-1}, s_t) = 1 \Leftrightarrow d_t = g(\overline{d}_{-1}, \widetilde{s}_t) \triangleq \widetilde{d}_{t,s_t}$; otherwise the value is zero, and $P(p_t|\overline{p}_{-1}, s_t)$ as the same. An update function $g$ is used to derive the updated symptom representation $\widetilde{s}_t$ and the current symptom-affected representations $\widetilde{d}_{t,s_t}$ and $\widetilde{p}_{t,s_t}$. The eventual probability can be reduced to the product of the weighted sum of two scoring functions. The sum with the scoring function $f$ is exactly the same as Equation (1), modeling the historical health conditions like a single visit. The current drug prescribing that we actually focus on is determined by another scoring function $\widetilde{f}$. Different from $f$ considering a single visit, the function $\widetilde{f}$ can leverage historical information since it is also related to previous recommendation results $\overline{y}_{-1}$. Each symptom $s'_t$ is sampled from the symptom sample set $\hat{\mathcal{S}}$ through the conditional distribution $P(s'_t|\overline{s}'_{-1})$. The condition probability can be estimated as $\hat{P}(s'_t|\overline{s}'_{-1})$, the ratio of the co-occurrence frequency of $s'_t$ and $\overline{s}'_{-1}$ to the frequency of $s'_t$. We can eventually obtain the current drug recommendation results $\widehat{\mathbf{y}}_t$ by estimating $\widetilde{f}$ in Equation (3). The details of network implementation for estimation is available in Section 4.3.

**DrugRec-k.** We propose another solution that extends drug recommendation to multi-visit scenario. An intuitive assumption is that for all historical visits of a patient, the more recent visit has a more significant effect on current drug prescriptions. Therefore, DrugRec-k only models information from the past $k$ visits to make the drug prescription of current visit more accurate. This implies the need to characterize the connections between the $k$ historical causal graphs and the current causal graph. Taking the symptom variable as an example, for the current $t$-th visit, we consider the effect of the symptom representations $S_{t-k}, S_{t-k+1}, \cdots, S_{t-1}$ on $S_t$. When $k = 0$, the problem degenerates into the single-visit scenario. In order to model how the $k$ historical symptom representations affect the current symptom representation, we make connections between each of the $k$ historical causal graphs and the current causal graph. Specifically, the $k$ paths $S_{t-k} \to S_t, S_{t-k+1} \to S_t, \cdots, S_{t-1} \to S_t$ are added to connect the historical representations with the current representation. For other variables $D_t$, $P_t$ and $Y_t$ at time $t$, such $k$ paths need to be linked in the same way. Based on the experience in DrugRec-a, the multi-visit causal effect we consider here is the joint intervention probability $P(y_t, y_{t-1}, \cdots, y_{t-k}|do(s_t), do(s_{t-1}), \cdots, do(s_{t-k}), m)$. The causal effect in the overall causal graph connected by $k + 1$ graphs can still be identifiable without breaking the conditions of front-door criterion, which is given by

$$P(y_t, y_{t-1}, \cdots, y_{t-k}|do(s_t), do(s_{t-1}), \cdots, do(s_{t-k}), m)$$

$$= \prod_{\kappa=1}^{k} \sum_{s'_{t-\kappa} \in \mathcal{S}} f\left(s'_{t-\kappa}, r(s_{t-\kappa}, d_{t-\kappa,s_{t-\kappa}}, p_{t-\kappa,s_{t-\kappa}}), m\right) P\left(s'_{t-\kappa}\right)$$

$$\cdot \sum_{r_t \in \mathcal{R}} \sum_{d_t \in \mathcal{D}} \sum_{p_t \in \mathcal{P}} P(d_t|d_{t-1}, \cdots, d_{t-k}, s_t) P(p_t|p_{t-1}, \cdots, p_{t-k}, s_t) P(r_t|s_t, d_t, p_t)$$

$$\cdot \sum_{s'_t \in \mathcal{S}} P\left(y_t|y_{t-1}, \cdots, y_{t-k}, s'_t, r_t, m\right) P\left(s'_t|s'_{t-1}, \cdots, s'_{t-k}\right). \tag{4}$$

We rearrange the Equation (4) and decompose it into the following two parts, corresponding to the patient's historical and current visit.

$$P(y_t, y_{t-1}, \cdots, y_{t-k}|do(s_t), do(s_{t-1}), \cdots, do(s_{t-k}), m)$$

$$\triangleq \prod_{\kappa=1}^{k} \sum_{s'_{t-\kappa} \in \mathcal{S}} f\left(s'_{t-\kappa}, r(s_{t-\kappa}, d_{t-\kappa,s_{t-\kappa}}, p_{t-\kappa,s_{t-\kappa}}), m\right) P\left(s'_{t-\kappa}\right)$$

$$\sum_{s'_t \in \mathcal{S}} \widetilde{f}\left(\widetilde{y}_{t-k}, s'_t, r\left(\widetilde{s}_t, \widetilde{d}_{t,s_t}, \widetilde{p}_{t,s_t}\right), m\right) P\left(s'_t|s'_{t-1}, \cdots, s'_{t-k}\right), \tag{5}$$

where $P(d_t|d_{t-1}, \cdots, d_{t-k}, s_t) = 1 \Leftrightarrow d_t = g'(d_{t-1}, \cdots, d_{t-k}, \widetilde{s}_t) \triangleq \widetilde{d}_{t,s_t}$; otherwise the value is zero, and $P(p_t|p_{t-1}, \cdots, p_{t-k}, s_t)$ as the same. The updated symptom representations $\widetilde{s}_t$ and current symptom-affected representations $\widetilde{d}_{t,s_t}$ and $\widetilde{p}_{t,s_t}$ can be derived by an update function $g'$. Here the historical and current recommendation decisions are also decomposable. The historical health conditions are modeled like $k$ single-visit scenarios. While for the current visit, the scoring function $\widetilde{f}$ is based on the aggregation of previous recommendation results $\widetilde{y}_{t-k} = agg(y_{t-1}, \cdots, y_{t-k})$, slightly different from DrugRec-a. In addition, the $s'_t$ is sampled by the estimated conditional

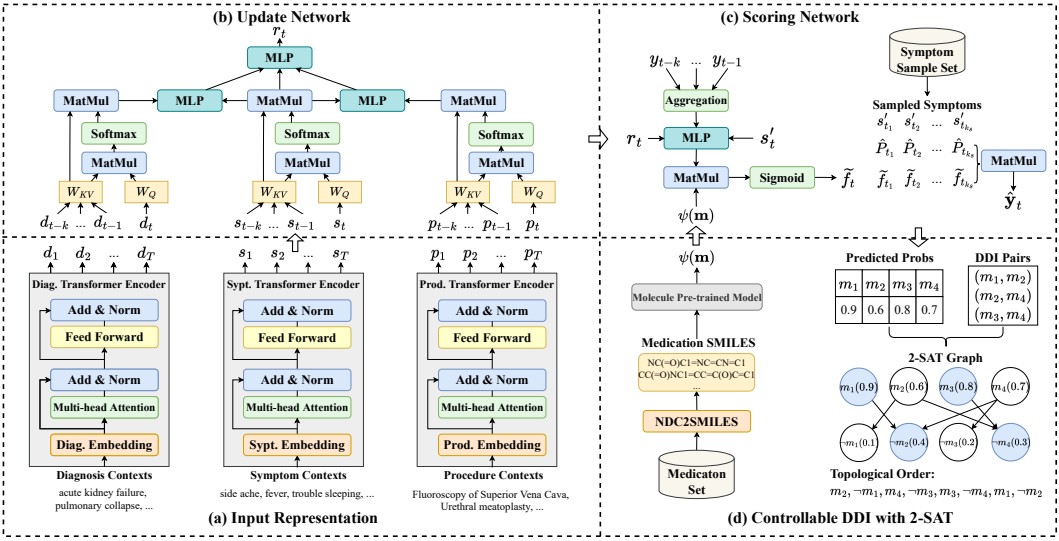

Figure 2: The model overview of DrugRec-k with the four modules : (a) Input Representation; (b) Update Network; (c) Scoring Network; and (d) Controllable DDI with 2-SAT.

distribution $\hat{P}(s'_t|s'_{t-1}, \cdots, s'_{t-k})$, which is the ratio of the corresponding co-occurrence frequency to the frequency of $s'_t$. The current drug recommendation results $\hat{\mathbf{y}}_t$ can be obtained by estimating $\widetilde{f}$ in Equation (5). The implementation details for estimation are available in Section 4.3.

## 4.2 Controllable DDI with 2-SAT

Another important issue in drug recommendation is coordinating multiple drugs, which differs from general recommendation systems. Significantly, the side effects of recommended drug combinations can threaten patient safety. Based on the information of paired drugs with harmful interactions stored in the DDI graph, previous methods mainly penalize the recommendation of DDI pairs by a DDI-related loss. However, there is no mature solution to handle DDIs at the inference stage. Here we propose a method that models the DDI as propositional satisfiability (SAT) problem. Specifically, we regard whether or not to retain a drug $i$ in the candidate set as a boolean variable that $m_i$. Then the DDI beetween drug $i$ and $j$ is represented as $(\neg m_i \vee \neg m_j)$. The DDI SAT problem is formulated by the conjunctions of all such clauses based on the drugs recommended by the causal graphical model. Taking the Figure 2(d) as an example, $m_1$ represents to keep the first drug, while $\neg m_1$ represents not to keep it, $m_2$, $m_3$, and $m_4$ as the same. The DDIs limit the coexistence of certain drugs, which can be formulated as $(\neg m_1 \vee \neg m_2) \wedge (\neg m_2 \vee \neg m_4) \wedge (\neg m_3 \vee \neg m_4)$.

Since every clause only has two literals, this is a 2-SAT problem that can be solved in polynomial time. Following [10], we first convert $\neg m_i \vee \neg m_j$ to the implicative normal form $m_i \Rightarrow \neg m_j, m_j \Rightarrow \neg m_i$. Then we construct a directed graph with two nodes $m_i$ and $\neg m_i$ for each drug $i$ and add the edges corresponding to the implications. Next, we find all strongly connected components (SCCs) of this graph using Tarjan's Algorithm [19], and then sort these SCCs in topological order. Although we can find a solution with any topological order, finding the best order is still an NP-hard problem. Therefore, we propose a heuristic method that enhance the topological sorting with the recommendation probability. When sorting the SCCs, we always select the SCC with no in-bond and the lowest probability, which is $P(m_i)$ for $m_i$ and $1 - P(m_i)$ for $\neg m_i$. This modification is critical for our method because the trivial solution (i.e., recommend no drugs) always exists, and our goal is to find a solution with a more significant likelihood. The rest step is identical with [10] that selecting $m_i$ or $\neg m_i$ by the one that comes later in the sorted SCCs. Since we let the nodes with lower probability come first, we are more likely to choose the drugs with a higher probability.

### 4.3 Model Implementation

In this section, we implement the the update function and scoring function through the neural network and introduce the training objective of our model. The model overview of DrugRec-k is illustrated in Figure 2. The architecture of DrugRec-a is similar, and a few differences will be described later.

**Input Representation.** For the drugs, we first obtain the SMILES[2] string of each drug in the data preprocessing. Next, we extract all the drug representations with the molecule pre-trained transformer model [31] and freeze the representation matrix as $\psi(\mathbf{m})$. For the diagnosis and procedure, previous work used multi-hot embedding of ICD-9 or ICD-10 codes[3], without guarantee that similar diagnoses and procedures will have similar representations. Instead, we convert the ICD codes to corresponding text descriptions in the dictionary tables and extract the symptoms from the clinical notes. Then we employ two-layer transformer encoders to encode the symptom, diagnosis, and procedure texts. We denote the output representations of transformer encoders as $s_t$, $d_t$ and $p_t$ at time $t$. Next, for the multi-visit scenario, the implementation for the two modeling schemes is slightly different on update network and scoring network.

**Update Network.** DrugRec-k employs an attention mechanism followed by subsequent Multilayer Perceptrons (MLPs) as three separate update functions $g_s$, $g_d$ and $g_p$. The updated representations $\widetilde{s}_t$, $\widetilde{d}_{t,s_t}$ and $\widetilde{p}_{t,s_t}$ can be formulated as Equations (6)-(8). The implementation detail of attention mechanism can be shown in Appendix B.

$$\widetilde{s}_t = g_s(s_{t-1}, \cdots, s_{t-k}) = \text{Attn}(s_t, s_{t-1}, \cdots, s_{t-k})_{s_t}, \tag{6}$$

$$\widetilde{d}_{t,s_t} = g_d(d_{t-1}, \cdots, d_{t-k}, \widetilde{s}_t) = \text{MLP}([\text{Attn}(d_t, d_{t-1}, \cdots, d_{t-k})_{d_t}, \widetilde{s}_t]), \tag{7}$$

$$\widetilde{p}_{t,s_t} = g_p(p_{t-1}, \cdots, p_{t-k}, \widetilde{s}_t) = \text{MLP}([\text{Attn}(p_t, p_{t-1}, \cdots, p_{t-k})_{p_t}, \widetilde{s}_t]). \tag{8}$$

While DrugRec-a uses an average aggregate function for the past $t-1$ representations to obtain $\overline{s}_{t-1}$, $\overline{d}_{t-1}$, $\overline{p}_{t-1}$, $\overline{y}_{t-1}$. Then its updated representations $\widetilde{s}_t$, $\widetilde{d}_{t,s_t}$ and $\widetilde{p}_{t,s_t}$ can be derived by three new update functions $g'_s$, $g'_d$ and $g'_p$, as shown in Equations (9)-(11).

$$\widetilde{s}_t = g'_s(\overline{s}_{t-1}) = \text{Attn}(s_t, \overline{s}_{t-1})_{s_t}, \tag{9}$$

$$\widetilde{d}_{t,s_t} = g'_d(\overline{d}_{t-1}, \widetilde{s}_t) = \text{MLP}([\text{Attn}(d_t, \overline{d}_{t-1})_{d_t}, \widetilde{s}_t]), \tag{10}$$

$$\widetilde{p}_{t,s_t} = g'_p(\overline{p}_{t-1}, \widetilde{s}_t) = \text{MLP}([\text{Attn}(p_t, \overline{p}_{t-1})_{p_t}, \widetilde{s}_t]). \tag{11}$$

Next, we concatenate the above representations and use a two-layer MLP to obtain the current patient visit representations as $r_t = \text{MLP}([\widetilde{s}_t, \widetilde{d}_{t,s_t}, \widetilde{p}_{t,s_t}])$.

**Scoring Network.** For the current $t$-th visit, we conduct a sample set with $k_s$ symptoms according to the estimation of the corresponding conditional probability mentioned in Section 4.1, denoted as $\hat{\mathcal{S}}_t$. For each sampled symptom representation $s'_t \in \hat{\mathcal{S}}_t$, a two-layer MLP is introduced as scoring functions $\widetilde{f}$. Together with previously obtained drug representation matrix $\psi(\mathbf{m})$, the patient visit representation $r_t$ and previous recommendation results, the final output of scoring function $\widetilde{f}_t$ is given by

$$\widetilde{f}_t = \begin{cases} \sigma(\text{MLP}([\overline{\mathbf{y}}_{t-1}, s'_t, r_t]) \cdot \psi(\mathbf{m})), & \text{for DrugRec-a,} \\ \sigma(\text{MLP}([agg(\mathbf{y}_{t-k}, \cdots, \mathbf{y}_{t-1}), s'_t, r_t]) \cdot \psi(\mathbf{m})), & \text{for DrugRec-k.} \end{cases} \tag{12}$$

where $\sigma(\cdot)$ represents the sigmoid function, and $agg$ represents the average aggregation. Therefore, the predicted recommendation probabilities $\hat{\mathbf{y}}_\mathbf{t} \in [0, 1]^{|\mathcal{M}|}$ can be formulated as Equation (13), the weighted sum for $\widetilde{f}_t$.

$$\widehat{\mathbf{y}}_t = \begin{cases} \displaystyle\sum_{s'_t \in \hat{\mathcal{S}}_t} \widetilde{f}_t \cdot \hat{P}(s'_t | \overline{s}_{t-1}), & \text{for DrugRec-a,} \\ \displaystyle\sum_{s'_t \in \hat{\mathcal{S}}_t} \widetilde{f}_t \cdot \hat{P}\left(s'_t | s'_{t-1}, \cdots, s'_{t-k}\right), & \text{for DrugRec-k.} \end{cases} \tag{13}$$

---

[2]Simplified molecular input line entry specification [24].

[3]A code list for the international classification of diseases (ICD), ninth or tenth revision [4].

**Training Objective.** During the training process, we divide all the medications into positive and negative cases according to ground truth in records. Correspondingly, we can obtain $\widehat{\mathbf{y}}_t^{(+)}$ and $\widehat{\mathbf{y}}_t^{(-)}$. Based on the causal theory, we maximize the average treatment effect (ATE) between the intervention with the observed symptoms $do(s_t)$ and the intervention without any symptoms $do(s_0)$. The ATE measures the difference in average outcomes between the observed symptoms and no symptom. We replace the $s_t$ with $s_0 = \mathbf{0}$, and then recalculate the updated representations and scoring outputs by Equations (6)-(13), resulting in $\widehat{\mathbf{y}}_0^{(+)}$. Since the outcome variable $Y$ follows a Bernoulli distribution, we can estimate the ATE among the positive cases as $\tau_t = \widehat{\mathbf{y}}_t^{(+)} - \widehat{\mathbf{y}}_0^{(+)}$. For all the $N$ patients in the records, the ATE-derived loss function can be formulated as

$$\mathcal{L}_{ate} = -\sum_{j=1}^{N}\sum_{t=1}^{T_j} \log\left(\sigma(\tau_t^{(j)})\right) \tag{14}$$

We also employ the following loss functions used in such previous works [27, 28]. We compute the binary cross-entropy (BCE) loss $\mathcal{L}_{bce}$ between prediction and ground-truth $\mathbf{m}_t^{(j)}$. The multi-label hinge loss $\mathcal{L}_{mul}$ is also introduced to ensure that the difference between the predicted probabilities of positive samples and those of a negative samples is at least 1 margin. In order to control the safety of the drug combination, the occurrence of DDIs needs to be penalized by the DDI loss $\mathcal{L}_{ddi}$.

In addition to the DDI loss, we hope to improve the accuracy of drug pair recommendations for better multiple drug coordination. Thus, we expand the single drug label $\mathbf{m}_t^{(j)} \in \{0,1\}^{|\mathcal{M}|}$ to the drug pair label $\widetilde{\mathbf{m}}_t^{(j)} \in \{0,1\}^{\frac{|\mathcal{M}|(|\mathcal{M}|-1)}{2}}$ and add the BCE loss for the drug pair $\mathcal{L}_{pair}$. The equations of the above four loss functions are available in Appendix B. The overall training loss is then formulated as Equation (15), where $\omega_{ate}, \omega_{mul}, \omega_{pair}, \omega_{ddi}, \gamma$ are the hyper-parameters for controlling the loss weight. We also set a DDI acceptance rate $\gamma$ to compare with the current DDI rate to determine whether to include $\mathcal{L}_{ddi}$ in the loss. The training process is shown in Algorithm 1 in Appendix B.

$$\mathcal{L} = \begin{cases} \mathcal{L}_{bce} + \omega_{ate}\mathcal{L}_{ate} + \omega_{mul}\mathcal{L}_{mul} + +\omega_{pair}\mathcal{L}_{pair} + \omega_{ddi}\mathcal{L}_{ddi}, & DDI > \gamma \\ \mathcal{L}_{bce} + \omega_{ate}\mathcal{L}_{ate} + \omega_{mul}\mathcal{L}_{mul} + +\omega_{pair}\mathcal{L}_{pair}, & DDI \leq \gamma \end{cases} \tag{15}$$

## 5  Experiments

**Data Processing.** We use the real-world EHRs obtained from publicly available MIMIC-III [9] and MIMIC-IV [8]. The DDI relations are obtained from TWOSIDES [20], and we convert the drug coding from NDC to ATC third level for integration with MIMIC. In addition to following the data processing methods of [28], we extract textual contents for the diagnosis and procedure by converting their ICD codes to corresponding text descriptions in the dictionary tables. We also obtain the symptom information from clinical notes. We further remove those patients who only had a single visit record or failed to extract any symptoms.[4] The datasets statistics are in Appendix C.

**Baselines and Metrics.** Under the exactly same data splits, we compare our method with the mainstream baselines. The baselines include instance-based methods: LR, ECC [14], LEAP [30] and longitudinal-based methods: RETAIN [2], GAMENet [17], MICRON [27], SafeDrug [28], COGNet [25] . Following the previous works [28, 25], we measure the model with standard effectiveness metrics: Jaccard Similarity Score (Jaccard), Precision Recall AUC (PRAUC) and F1 score (F1). DDI Rate and number of drugs are also included, lower is better. The baseline and metric details are available in Appendix C.

**Main Results.** Table 1 show the experimental results of all methods on MIMIC-III and MIMIC-IV. Instance-based methods (LR, ECC and LEAP) perform poorly without considering historical records. RETAIN and GAMENet outperform instance-based methods on the effectiveness metrics, yet improve DDI Rate exceeding the ground-truth. The DDI Rate of MICRON and SafeDrug is lower than other baselines, as they are specially designed with controllable DDI loss. COGNet performs best among baselines in terms of effectiveness metrics, but its number of recommended drugs is too large, and the DDI Rate is also a bit high. Our DrugRec outperforms all the baselines, where DrugRec-k performs better than DrugRec-a. This means that simply compressing the representation of all previous visits cannot obtain the best performance. The highest effectiveness metrics show that

---

[4]The code and data are available at https://github.com/ssshddd/DrugRec.

Table 1: Experimental results on MIMIC-III / MIMIC-IV. Ground-truth DDI Rate is 0.0754 / 0.0713.

| Model | Jaccard | F1 | PRAUC | DDI Rate | Avg. # of Drugs |
|---|---|---|---|---|---|
| LR | 0.4896 / 0.3844 | 0.6491 / 0.5379 | 0.7568 / 0.6568 | 0.0774 / 0.0645 | 17.4894 / 9.3892 |
| ECC | 0.4799 / 0.3680 | 0.6390 / 0.5173 | 0.7572 / 0.6541 | 0.0760 / 0.0648 | 16.8464 / 8.7081 |
| LEAP | 0.4465 / 0.3653 | 0.6097 / 0.5201 | 0.6490 / 0.5314 | 0.0657 / 0.0570 | 19.0166 / 12.5083 |
| RETAIN | 0.4780 / 0.3903 | 0.6397 / 0.5471 | 0.7601 / 0.6563 | 0.0814 / 0.0618 | 18.9820 / 10.0435 |
| GAMENet | 0.5039 / 0.3957 | 0.6609 / 0.5525 | 0.7632 / 0.6479 | 0.0832 / 0.0757 | 26.1520 / 17.5848 |
| MICRON | 0.5076 / 0.4009 | 0.6634 / 0.5545 | 0.7685 / 0.6584 | 0.0612 / 0.0605 | 18.0141 / 11.1404 |
| SafeDrug | 0.5090 / 0.4082 | 0.6664 / 0.5651 | 0.7647 / 0.6495 | 0.0658 / 0.0553 | 20.4243 / 12.6161 |
| COGNet | 0.5134 / 0.4131 | 0.6706 / 0.5660 | 0.7677 / 0.6460 | 0.0784 / 0.0596 | 24.1675 / 19.3966 |
| **DrugRec-a** | 0.5196 / 0.4162 | 0.6756 / 0.5690 | 0.7680 / 0.6507 | 0.0606 / 0.0401 | 23.7549 / 13.7241 |
| **DrugRec-k** | **0.5220 / 0.4194** | **0.6771 / 0.5713** | **0.7720 / 0.6558** | **0.0597 / 0.0396** | 22.0006 / 13.4880 |

Table 2: Ablation study for DrugRec on MIMIC-III dataset.

| Model | Jaccard | F1 | PRAUC | DDI Rate | Avg. # of Drugs |
|---|---|---|---|---|---|
| DrugRec (w/o sypt.) | $0.5092 \pm 0.0044$ | $0.6662 \pm 0.0040$ | $0.7612 \pm 0.0042$ | $0.0600 \pm 0.0009$ | $21.9129 \pm 0.1289$ |
| DrugRec (w/o $\mathcal{L}_{ate}$) | $0.5142 \pm 0.0041$ | $0.6710 \pm 0.0036$ | $0.7673 \pm 0.0041$ | $0.0597 \pm 0.0006$ | $21.8849 \pm 0.1655$ |
| DrugRec (single) | $0.5167 \pm 0.0044$ | $0.6730 \pm 0.0039$ | $0.7667 \pm 0.0040$ | $0.0600 \pm 0.0006$ | $22.2260 \pm 0.1243$ |
| DrugRec (w/o sat) | $0.5240 \pm 0.0034$ | $0.6787 \pm 0.0035$ | $0.7674 \pm 0.0045$ | $0.0774 \pm 0.0008$ | $23.9584 \pm 0.2013$ |
| **DrugRec** | $\mathbf{0.5220 \pm 0.0034}$ | $\mathbf{0.6771 \pm 0.0031}$ | $\mathbf{0.7720 \pm 0.0036}$ | $\mathbf{0.0597 \pm 0.0006}$ | $22.0006 \pm 0.1604$ |

our multi-visit causal drug recommendation can indeed make drug recommendation more accurate, and the lowest average DDI Rate indicates that our controllable DDI with 2-SAT works. To test the significance and robustness of our metrics, we conduct a two-sample T-test on each metric between our method and each baseline. The results in Table 1 are obtained through 10 rounds of bootstrapping sampling on the test set, and the standard deviation with $p$-values are reported in Appendix C. Our method performs best significantly on both effectiveness metrics and DDI Rate, with all $p$-values below 0.01 for each metric in each baseline.

**Ablation Study.** In order to verify the effectiveness of the individual components (symptom information, causal view for drug recommendation, multi-visit causal drug recommendation and controllable DDI with 2-SAT ) in our model, we design different model variants for ablation analysis in Table 2. We have the following four model variants. 1) **DrugRec (w/o sypt.)** is implemented without applying any symptom information. This general discriminative model is equivalent to removing both $C$ and $S$ from the graph in Figure 1, and its performance is the poorest in comparison. 2) When we add the symptom information to DrugRec (w/o sypt.) and obtain **DrugRec (w/o $\mathcal{L}_{ate}$)**, all the metric results are better than before. It indicates that utilizing the necessary symptom information will significantly help the effectiveness of drug recommendations. 3) **DrugRec (single)** models the single-visit causal effect in the recommendation with the causal view of drug recommendation. The metric results of Jaccard and F1 have improved compared to the previous model. However, compared to the full model, DrugRec (single) performs worse on all metrics since it ignores the patient's historical health condition. Overall, it demonstrates the effectiveness of the causal view for drug recommendation and multi-visit causal drug recommendation. 4) **DrugRec (w/o sat)** removes the module of controllable DDI with 2-SAT. The DDI Rate and the number of drugs are significantly larger than all the model variants. This confirms that our 2-SAT algorithm is indeed effective for controlling DDI Rate.

**Multi-Visit Analysis for DrugRec-k.** To further investigate the impact of the different number of past visits for DrugRec-k, we conduct a comparative experiment with different values of $k$ on MIMIC-III. We consider the case where $k$ takes values from 0 to 3, and $k = 0$ corresponds to the single-visit scenario. Table 3 shows the results of effectiveness metrics for different $k$. The model performance of DrugRec-k is optimal when $k = 2$. All effectiveness metrics increase as $k$ increases from 0 to 2, and drop a little when $k = 3$. Since the value of $k$ is related to the assumption of the impact of the patient's past health condition on the current drug prescribing, we analyze the above results from two perspectives. On the one hand, it suggests that too old information recorded in the historical visits may not be instructive or even misleading for the current prescribing. The most useful

historical information is in the last two visits. On the other hand, the average visits for all patients in MIMIC-III is 2.59, with very few patients having over 3 visits. This also shows that the $k$ should not be too high.

For further analysis and discussion, some supplementary experiments are also performed, which are detailed in Appendix D.

Table 3: Study of different $k$ for DrugRec-k.

| $k$ | Jacarrd | F1 | PRAUC |
|---|---|---|---|
| 0 | 0.5167 | 0.6730 | 0.7667 |
| 1 | 0.5203 | 0.6764 | 0.7685 |
| 2 | **0.5220** | **0.6771** | **0.7720** |
| 3 | 0.5210 | 0.6765 | 0.7711 |

## 6 Conclusion

This work focuses on three essential aspects of the drug recommendation task and proposes novel models to enhance them. We first design a causal graphical model for the drug recommendation task and deconfound the effect of the invisible recommendation bias with front-door adjustment. To better model a patient's historical health condition, we further characterize two modeling schemes that extend the causal graph to the multi-visit scenario. We also propose a novel 2-SAT algorithm to coordinate multiple drugs with DDIs. Comprehensive experimental results on MIMIC-III/IV datasets show that our method significantly outperforms all baselines on the effectiveness metrics and successfully controls harmful DDI relations. We hope this method could help improve the efficiency and accuracy of the real-world drug prescribing process in hospital and benefit the public health as well as well-being.

## Acknowledgements

We would like to thank the anonymous reviewers for their insightful comments. This work was supported by National Natural Science Foundation of China (NSFC Grant No. 62122089 and No. 61876196), Beijing Outstanding Young Scientist Program NO. BJJWZYJH012019100020098, and Intelligent Social Governance Platform, Major Innovation & Planning Interdisciplinary Platform for the "Double-First Class" Initiative, Renmin University of China. We also wish to acknowledge the support provided and contribution made by Public Policy and Decision-making Research Lab of RUC. Rui Yan is supported by Beijing Academy of Artificial Intelligence (BAAI).

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
