# Appendix of Debiased, Longitudinal and Coordinated Drug Recommendation through Multi-Visit Clinic Records

**Hongda Sun, Shufang Xie, Shuqi Li, Yuhan Chen, Ji-Rong Wen, Rui Yan**[*]
Gaoling School of Artificial Intelligence, Renmin University of China, Beijing, China
{sunhongda98,shufangxie,shuqili,yuhanchen,jrwen,ruiyan}@ruc.edu.cn

## A   Details of Causal Analysis

We need two steps to analyze causal effects between two variables. First, we need to design a complete causal graph that conforms to the domain knowledge. Second, we need to determine whether the causal effect between our variables of interest is identifiable.

For example, in Figure 1(a), we denote $T$ as the *treatment* variable and $Y$ as the *outcome* variable. We want to estimate the causal effect of $T$ on $Y$. There is a *confounder* $C$ in the causal graph, which is a common cause of $T$ and $Y$. In other words, there is a *backdoor path* $T \leftarrow C \rightarrow Y$ between $T$ and $Y$, and $C$ blocks the path. Correspondingly, $T \rightarrow W \rightarrow Y$ is called the *front-door path*. In order to represent the causal effect of $T$ on $Y$, the intervention probability $P(y|do(t))$ is introduced to replace the original conditional probability $P(y|t)$, which means the probability of $Y = y$ when cutting off the path $C \rightarrow T$ and fixing $T = t$. How to remove the *do*-operator is thus a key problem to identify the causal effect of $T$ on $Y$. In this work, we leverage the front-door criterion [2] to solve the problem of identifiability since the confounder is unobservable. If the mediator $W$ satisfies the front-door criterion, then the causal effect of $T$ on $Y$ is identifiable and is given by the following front-door adjustment formula.

$$P(y|do(t)) = \sum_w P(w|t) \sum_{t'} P(y|t', w)P(t')$$

For the drug recommendation task, we propose a causal graph, as shown in Figure 1(b). We denote the symptom $(S)$ as the treatment variable and the drug recommendation probability $(Y)$ as the outcome variable. Also, $D$, $P$ and $R$ can be treated as mediators, which satifies the front-door criterion like Figure 1(a). Then the causal effect $P(y|do(s), m)$ is formulated as

$$\begin{aligned}
P(y|do(s), m) &= \sum_{r \in \mathcal{R}} \sum_{d \in \mathcal{D}} \sum_{p \in \mathcal{P}} P(d|s)P(p|s)P(r|s, d, p) \sum_{s' \in \mathcal{S}} P(y|s', r, m)P(s') \\
&= \sum_{r \in \mathcal{R}} P(r|s, d_s, p_s) \sum_{s' \in \mathcal{S}} P(y|s', r, m)P(s') \\
&= \sum_{s' \in \mathcal{S}} P(y|s', r(s, d_s, p_s), m)P(s') \\
&\triangleq \sum_{s' \in \mathcal{S}} f(s', r(s, d_s, p_s), m)P(s').
\end{aligned}$$

---

[*]Corresponding author: Rui Yan (ruiyan@ruc.edu.cn)

36th Conference on Neural Information Processing Systems (NeurIPS 2022).

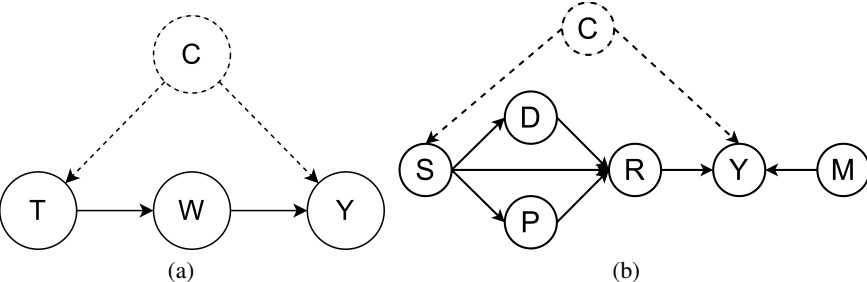

Figure 1: (a) A simple example for causal graph with unobservable confounder. The variables represent: C: Confounder, T: Treatment, W: Mediator, Y: Outcome. (b) The causal graph for drug recommendation. The variables represent: D: diagnosis, P: procedure, R: patient visit representation, S: symptom, M: medication, Y: recommend or not. The dotted arrows and circles represent unobservable variable and links.

## B    Model Details

### B.1    Attention for Update Network

By introducing the learnable transformation matrices of query $W_Q$, key $W_K$ and value $W_V$, the attention mechanism for vectors $x = (x_1, \cdots, x_L)$ is given by

$$Q_x, K_x, V_x = x\, W_Q, x\, W_K, x\, W_V,$$

$$\alpha_{m,n} = \frac{\exp(Q_{x_m} K_{x_n})}{\sum_{l=1}^{L} \exp(Q_{x_m} K_{x_l})},$$

$$\text{Attn}(x_1, \cdots, x_L)_{x_m} = \sum_{n=1}^{L} \alpha_{m,n} V_{x_n}.$$

In particular, we select a shared-KV attetion scheme in our update network by sharing the matrices of $W_Q$ and $W_K$ as $W_{KV}$.

### B.2    General Loss

In addition to our original $\mathcal{L}_{ate}$, the ohter four loss functions $\mathcal{L}_{bce}$, $\mathcal{L}_{mul}$, $\mathcal{L}_{pair}$ and $\mathcal{L}_{ddi}$ are formulated as follows.

$$\mathcal{L}_{bce} = -\sum_{j=1}^{N}\sum_{t=1}^{T_j} \mathbf{m}_t^{(j)} \log(\widehat{\mathbf{y}}_t^{(j)}) + (1 - \mathbf{m}_t^{(j)}) \log(1 - \widehat{\mathbf{y}}_t^{(j)})$$

$$\mathcal{L}_{mul} = \sum_{j=1}^{N} \sum_{\substack{\hat{y}^{(+)} \in \widehat{\mathbf{y}}_t^{(j)(+)}, \\ \hat{y}^{(-)} \in \widehat{\mathbf{y}}_t^{(j)(-)}}} \frac{max(1 - (\hat{y}^{(+)} - \hat{y}^{(-)}), 0)}{|\mathcal{M}|}$$

$$\mathcal{L}_{ddi} = \sum_{j=1}^{N}\sum_{t=1}^{T_j}\sum_{u=1}^{|\mathcal{M}|}\sum_{v=1}^{|\mathcal{M}|} \mathbf{A}_{uv}\widehat{\mathbf{y}}_{t_u}^{(j)}\widehat{\mathbf{y}}_{t_v}^{(j)}$$

$$\mathcal{L}_{pair} = -\sum_{j=1}^{N}\sum_{t=1}^{T_j}\sum_{u=1}^{|\mathcal{M}|-1}\sum_{v=u+1}^{|\mathcal{M}|} \widetilde{\mathbf{m}}_{t_{u,v}}^{(j)} \log(\widehat{\mathbf{y}}_{t_u}^{(j)}\widehat{\mathbf{y}}_{t_v}^{(j)}) + (1 - \widetilde{\mathbf{m}}_{t_{u,v}}^{(j)}) \log(1 - \widehat{\mathbf{y}}_{t_u}^{(j)}\widehat{\mathbf{y}}_{t_v}^{(j)})$$

### B.3    Training Algorithm

Our DrugRec model is trained with the Algorithm 1. The equtaions mentioned below are all from the main text.

**Algorithm 1** Training Process for DrugRec
***

**Input**: EHR Training set: $\mathbf{X}_{tra} = \{\mathbf{X}^{(1)}, \ldots, \mathbf{X}^{(N_{tra})}\}$, DDI adjacency matrix: $\mathbf{A}$, the number of sampled symptoms: $k_s$, training epoches: $L$, weights of loss function: $\omega_{ate}, \omega_{mul}, \omega_{pair}, \omega_{ddi}, \gamma$.
**Parameter**: Learnable parameters in networks.
**Output**: The recommendation results for all patients $\widehat{\mathbf{y}}$.

1: **for** $i = 1$ to $L$ **do**
2:    **for** $j = 1$ to $N_{tra}$ **do**
3:       $\mathcal{L} \leftarrow 0$.
4:       **for** $t = 1$ to $T_j$ **do**
5:          Calculate $\widetilde{s}_t^{(j)}, \widetilde{d}_{t,s_t}^{(j)}, \widetilde{p}_{t,s_t}^{(j)}$ using Eq.(4)-(9) in the main text.
6:          Calculate $r_t^{(j)} = \mathrm{MLP}([\widetilde{s}_t^{(j)}, \widetilde{d}_{t,s_t}^{(j)}, \widetilde{p}_{t,s_t}^{(j)}])$.
7:          $\hat{\mathcal{S}}_t^{(j)} \leftarrow K$ random symptoms from the estimated conditional probability.
8:          Calculate $\widehat{\mathbf{y}}_t^{(j)}$ using Eq.(11).
9:          Recalculate $\widehat{\mathbf{y}}_0^{(j)}$ using Eq.(4)-(11) ($s_t \leftarrow s_0 = \mathbf{0}$).
10:         Calculate $\mathcal{L}_t^{(j)}$ using Eq.(13).
11:         $\mathcal{L} = \mathcal{L} + \mathcal{L}_t^{(j)}$.
12:       **end for**
13:       Update the learnable parameters.
14:    **end for**
15: **end for**
16: **return** $\widehat{\mathbf{y}}$.
***

# C   Experiment Details

## C.1   Datasets

After preprocessing real-world health records in MIMIC-III and MIMIC-IV, the statistics of the eventual datasets can be shown in Table 1. The distribution of number of visits in two MIMIC datasets are shown in Figure 2.

Table 1: The statistics of the eventual datasets.

| Items | MIMIC-III | MIMIC-IV |
|---|---|---|
| # of patients | 5208 | 6136 |
| # of clinical visits | 13490 | 17813 |
| sypt./ diag./ prod. / med. space size | 428/1895/1378/112 | 163/1851/4001/121 |
| avg. # of visits | 2.59 | 2.90 |
| avg. # of sypt./ diag./ prod./ med. per visit | 7.67/10.24/3.85/11.30 | 1.09/11.78/2.18/6.68 |
| total # of DDI pairs | 337 | 337 |

## C.2   Baseline Details

- **LR** is a logistic regression algorithm with L2 regularization, where the multi-hot diagnosis and procedure vector are concatenated as the input feature, and the One-vs-Rest classifier is used for multi-label classification.

- **ECC** [3] is a 10-member ensemble classifier chain for dependent series of multi-label classification.

- **LEAP** [8] is a typical instance-based method that treats the drug recommendation as a sentence generation.

- **RETAIN** [1] is makes sequential prediction of medication combination based on a two-level neural attention model.

- **GAMENet** [4] uses the memory neural network to store the information in historical health records and graph convolution network to encode the EHR and DDI graph.

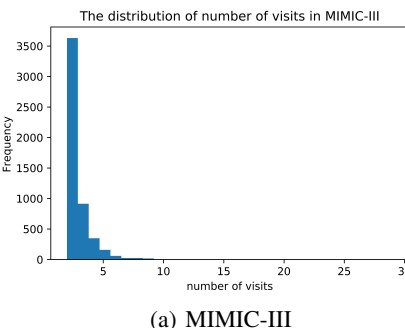

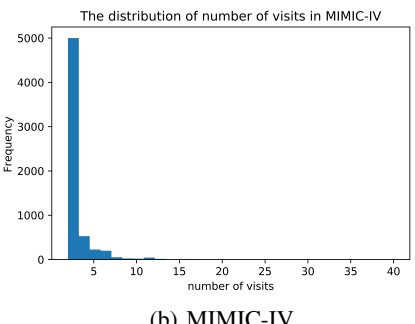

(a) MIMIC-III                    (b) MIMIC-IV

Figure 2: The distribution of number of visits in two MIMIC datasets.

- **MICRON** [6] proposes a recurrent residual learning model for predicting medication changes, which are used to reconstruct the recommended drugs.
- **SafeDrug** [7] leverages dual molecular encoders for rich molecule structures and uses DDI controller for safe recommendation.
- **COGNet** [5] introduces a novel copy-or-predict mechanism to regard drug recommendation as a sequence generation problem based on encoder-decoder framework.

## C.3 Metric Details

For the evaluation metrics, we measure models with standard effectiveness metrics: Jaccard Similarity Score (Jaccard), Precision Recall AUC (PRAUC) and F1 score (F1). We also meausure the safety with DDI Rate. For the patient $j$ at the $t$-th visit, we denote the groud-truth drug combination as $\mathbf{m}_t^{(j)}$, and the predicted drug combination as $\hat{\mathbf{m}}_t^{(j)}$.

We can formulate the average Jaccard for patient $j$ as

$$\text{Jaccard}_j = \frac{1}{T_j} \sum_{t=1}^{T_j} \frac{|\{i : \mathbf{m}_{t_i}^{(j)} = 1\} \cap \{i : \hat{\mathbf{m}}_{t_i}^{(j)} = 1\}|}{|\{i : \mathbf{m}_{t_i}^{(j)} = 1\} \cup \{i : \hat{\mathbf{m}}_{t_i}^{(j)} = 1\}|} \tag{1}$$

The average F1 for patient $j$ is formulated as

$$\text{F1}_j = \frac{1}{T_j} \sum_{t=1}^{T_j} \frac{2R_t^{(j)} P_t^{(j)}}{R_t^{(j)} + P_t^{(j)}} \tag{2}$$

where the recall and precision at each time $t$ for patient $j$ are formulated as

$$R_t^{(j)} = \frac{|\{i : \mathbf{m}_{t_i}^{(j)} = 1\} \cap \{i : \hat{\mathbf{m}}_{t_i}^{(j)} = 1\}|}{|\{i : \mathbf{m}_{t_i}^{(j)} = 1\}|} \tag{3}$$

$$P_t^{(j)} = \frac{|\{i : \mathbf{m}_{t_i}^{(j)} = 1\} \cap \{i : \hat{\mathbf{m}}_{t_i}^{(j)} = 1\}|}{|\{i : \hat{\mathbf{m}}_{t_i}^{(j)} = 1\}|} \tag{4}$$

The average PRAUC for patient $j$ can be calculated as

$$\text{PRAUC}_j = \frac{1}{T_j} \sum_{t=1}^{T_j} \sum_{m=1}^{|\mathcal{M}|} P_{t,m}^{(j)} (R_{t,m}^{(j)} - R_{t,m-1}^{(j)}) \tag{5}$$

where $P_{t,m}^{(j)}$ represents the precision at cut-off $m$ in the ordered list, and the change of recall from drug $m - 1$ to $m$ is the latter item.

The DDI Rate for patient $j$ is calculated as

$$\text{DDI}_j = \frac{\sum_{t=1}^{T_j} \sum_{l,k \in \{i:\hat{\mathbf{m}}_{t_i}^{(j)}=1\}} \mathcal{I}\{\mathbf{A}_{lk} = 1\}}{\sum_{t=1}^{T_j} \sum_{l,k \in \{i:\hat{\mathbf{m}}_{t_i}^{(j)}=1\}} 1} \tag{6}$$

where $\mathcal{I}$ is indicator function for counting the positions where the DDI matrix takes a value of 1.

## C.4  Implementation Details

We follow the same data split setting [7], dividing the dataset into training, validation and test set by different patients with a ratio of 4:1:1. The training process has 100 epoches using the Adam optimizer and linear warm-up cosine-annealing learning rate scheduler with the base learning rate of 5e-4. The number of layers for all transformer encoders and MLPs is 2. The embedding size and hidden size is 64. The number of sampled symptoms $k_s$ is set to 5. All dropout parameters are set to 0.1. The hyperparameters of the loss function $\omega_{ate}$, $\omega_{mul}$, $\omega_{pair}$, $\omega_{ddi}$ and $\gamma$ are set to 0.005, 0.1, 1.0, 0.5 and 0.05. We implement our experiments on two Nvidia A40 GPUs.

## C.5  Standard Deviations & $p$-values of Main Results

We provide results of the standard deviation and $p$-value results in MIMIC-III (Table 2) and MIMIC-IV (Table 3). We present standard deviations after $\pm$ and $p$-values for significant tests are in the parentheses.

Table 2: Experimental results on MIMIC-III.

| Model | Jaccard | F1 | PRAUC | DDI Rate |
|---|---|---|---|---|
| LR | $0.4896 \pm 0.0025$ (3e-15) | $0.6491 \pm 0.0024$ (1e-14) | $0.7568 \pm 0.0025$ (2e-9) | $0.0774 \pm 0.0012$ (2e-19) |
| ECC | $0.4799 \pm 0.0022$ (0.0) | $0.6390 \pm 0.0022$ (0.0) | $0.7572 \pm 0.0026$ (4e-9) | $0.0760 \pm 0.0010$ (8e-20) |
| LEAP | $0.4465 \pm 0.0037$ (0.0) | $0.6097 \pm 0.0036$ (0.0) | $0.6490 \pm 0.0033$ (0.0) | $0.0657 \pm 0.0010$ (3e-12) |
| RETAIN | $0.4780 \pm 0.0036$ (2e-16) | $0.6397 \pm 0.0036$ (2e-15) | $0.7601 \pm 0.0035$ (6e-7) | $0.0814 \pm 0.0018$ (3e-18) |
| GAMENet | $0.5039 \pm 0.0021$ (3e-11) | $0.6609 \pm 0.0020$ (5e-11) | $0.7632 \pm 0.0027$ (8e-6) | $0.0832 \pm 0.0005$ (9e-26) |
| MICRON | $0.5076 \pm 0.0037$ (4e-8) | $0.6634 \pm 0.0035$ (3e-8) | $0.7685 \pm 0.0038$ (0.047) | $0.0612 \pm 0.0008$ (2e-4) |
| SafeDrug | $0.5090 \pm 0.0038$ (2e-7) | $0.6664 \pm 0.0033$ (6e-7) | $0.7647 \pm 0.0020$ (3e-5) | $0.0658 \pm 0.0003$ (2e-16) |
| COGNet | $0.5134 \pm 0.0027$ (7e-6) | $0.6706 \pm 0.0043$ (1e-3) | $0.7677 \pm 0.0013$ (2e-3) | $0.0784 \pm 0.0005$ (5e-24) |
| **DrugRec** | $\mathbf{0.5220 \pm 0.0034}$ | $\mathbf{0.6771 \pm 0.0031}$ | $\mathbf{0.7720 \pm 0.0036}$ | $\mathbf{0.0597 \pm 0.0006}$ |

Table 3: Experimental results on MIMIC-IV.

| Model | Jaccard | F1 | PRAUC | DDI Rate |
|---|---|---|---|---|
| LR | $0.3844 \pm 0.0028$ (0.0) | $0.5379 \pm 0.0031$ (4e-16) | $0.6568 \pm 0.0036$ (5e-6) | $0.0645 \pm 0.0012$ (7e-22) |
| ECC | $0.3680 \pm 0.0041$ (0.0) | $0.5173 \pm 0.0047$ (0.0) | $0.6541 \pm 0.0030$ (3e-8) | $0.0648 \pm 0.0018$ (3e-19) |
| LEAP | $0.3653 \pm 0.0028$ (0.0) | $0.5201 \pm 0.0033$ (0.0) | $0.5314 \pm 0.0038$ (0.0) | $0.0570 \pm 0.0011$ (2e-19) |
| RETAIN | $0.3903 \pm 0.0038$ (3e-14) | $0.5471 \pm 0.0040$ (2e-12) | $0.6563 \pm 0.0055$ (1e-4) | $0.0618 \pm 0.0025$ (5e-16) |
| GAMENet | $0.3957 \pm 0.0035$ (3e-13) | $0.5525 \pm 0.0041$ (2e-10) | $0.6479 \pm 0.0055$ (3e-8) | $0.0757 \pm 0.0014$ (1e-23) |
| MICRON | $0.4009 \pm 0.0044$ (4e-10) | $0.5545 \pm 0.0048$ (8e-9) | $0.6584 \pm 0.0043$ (2e-4) | $0.0605 \pm 0.0017$ (3e-18) |
| SafeDrug | $0.4082 \pm 0.0026$ (3e-9) | $0.5651 \pm 0.0028$ (3e-5) | $0.6495 \pm 0.0036$ (9e-10) | $0.0553 \pm 0.0010$ (4e-19) |
| COGNet | $0.4131 \pm 0.0020$ (1e-6) | $0.5660 \pm 0.0019$ (2e-5) | $0.6460 \pm 0.0017$ (8e-14) | $0.0596 \pm 0.0005$ (9e-24) |
| **DrugRec** | $\mathbf{0.4194 \pm 0.0020}$ | $\mathbf{0.5713 \pm 0.0022}$ | $\mathbf{0.6658 \pm 0.0026}$ | $\mathbf{0.0396 \pm 0.0007}$ |

## D  Supplementary Experiments

### D.1  The Effect of Number of Visits

We stratifed the datasets based on different number of visits to study its impact on the performance of different models. Follwing the preprocessing scripts of [7], patients with only 1 visit were removed. Thus, we stratified the test set into 3 groups by the total number of patient visits: 2, 3 and more than 3 visits. The comparison of various methods on different number of visits is shown in Table 4. Here we chose the recent COGNet and SafeDrug as stronger baselines. Our DrugRec outperformed baselines on all metrics in each group. For different values of $k$ in DrugRec, DrugRec-2 has the best performance, which concurs with the observation in Section Experiments.

Table 4: The effect of number of visits.

| | Jaccard | | | PRAUC | | | F1 | | |
|---|---|---|---|---|---|---|---|---|---|
| | 2 | 3 | >3 | 2 | 3 | >3 | 2 | 3 | >3 |
| SafeDrug | 0.5114 | 0.5016 | 0.5006 | 0.7653 | 0.7652 | 0.7587 | 0.6678 | 0.6620 | 0.6617 |
| COGNet | 0.5131 | 0.5146 | 0.5140 | 0.7675 | 0.7689 | 0.7674 | 0.6703 | 0.6722 | 0.6705 |
| DrugRec-1 | 0.5216 | 0.5170 | 0.5146 | 0.7687 | 0.7681 | 0.7674 | 0.6775 | 0.6731 | 0.6724 |
| DrugRec-2 | **0.5237** | **0.5176** | **0.5147** | **0.7729** | **0.7691** | **0.7690** | **0.6782** | **0.6739** | 0.6729 |
| DrugRec-3 | 0.5225 | 0.5170 | **0.5147** | 0.7720 | 0.7683 | 0.7680 | 0.6774 | 0.6738 | **0.6732** |

## D.2 Case Study

To verify the impact of considering the existence of the hidden confounder for multi-visit drug recommendation, an example recommendation drugs for a patient with three visits is provided in Table 5. We choose COGNet and our ablation model DrugRec (w/o $\mathcal{L}_{ate}$) as stronger baselines ignoring the hidden confounder. Here we detail the patient's symptoms and ICD codes of diagnoses and procedures for each visit. At the same time, ground-truth medications prescribed by doctors and recommended medications by different methods are also included. During the several visits, the patient's disease condition has changed, and the ground-truth medications have also changed accordingly. In the patient's early visit (1st), the performance of the baselines and DrugRec is similar. While in later visits (2nd, 3rd), the baselines may accumulate recommendation bias. DrugRec can alleviate this problem due to modeling the hidden confounder.

For COGNet and DrugRec (w/o $\mathcal{L}_{ate}$) ignoring the hidden confounder, they will always predict a drug "J01M" that is frequently recommended in the training set, but is not included in the current ground truth. This makes drug recommendations more biased. Since DrugRec models the hidden confounder, it will not always recommend the wrong drug and achieve debiased drug recommendations, improving the recommendation accuracy.

Table 5: Example recommended drugs for a patient with three visits. Here "FN" refers to the number of drugs that are in ground-truth but not predicted, while "FP" indicates the number of drugs predicted but not in ground-truth. The key drugs that cause the prediction bias are marked in bold and are always incorrectly recommended in this case.

| Patient Disease Condition | Method | Recommended Drugs (ATC3) |
|---|---|---|
| 1st Visit
Diag.: 2724, 4280, 41401, 25000, 4439, V4582, 5180, 3004, 4240, E8781, 79001, 28860, V103, 42823, E8889, V4364, 41072, 92232, 40291
Prod: 9390
Sypt.: cough, shortness of breath, chest pain, nausea, constipation | Ground Truth | N02B, A01A, A02B, A06A, B05C, C07A, C03C, A12B, N02A, N06A, B01A, C10A, C01B, N05C, C09A, H04A |
| | COGNet | N02B, A01A, A02B, A06A, B05C, A12C, C07A, C03C, N02A, **J01M**, B01A, C10A, N05C, C09A, R03A, R01A, G04C (TP=12, FN=4, FP=5) |
| | DrugRec (w/o Late) | N02B, A01A, A02B, A06A, B05C, A10A, C07A, A12B, N02A, **J01M**, B01A, C01B, C09A, A07D, D04A (TP=11, FN=5, FP=4) |
| | DrugRec | N02B, A01A, A02B, A06A, B05C, C07A, C03C, A12B, N02A, N06A, B01A, C10A, C09A, C01D (TP=13, FN=3, FP=1) |
| 2nd Visit
Diag.: 78551, 2724, 45829, 4280, 41401, 42731, 4271, 4019, V4582, 99672, 3004, 4240, 4168, 9971, 412, 5845, 79001, E8497, E8790, 42823, 4260, 71590, E8782, E9444, 45989
Prod: 3893, 0066, 3895, 3995, 8856, 3722, 8964, 8842, 0041, 3607, 0046, 3964, 3768
Sypt.: cough, shortness of breath, bleeding, depression, chills, sob | Ground Truth | N02B, A01A, A06A, B05C, A12A, A12C, C01C, A07A, C07A, C03C, A12B, N02A, N06A, B01A, A03B, C10A, C01B, N05C, C09A, B02B, C01D, N05B, R05C, R01A, D04A, C03B |
| | COGNet | N02B, A01A, A06A, B05C, A12A, A07A, A10A, N01A, C07A, C03C, A12B, N07A, C02D, N02A, **J01M**, B01A, A03B, C10A, C01B, N05C, C09A, C08C, C01D, A04A, D11A (TP=16, FN=10, FP=9) |
| | DrugRec (w/o Late) | N02B, A01A, A06A, B05C, A12A, C01C, A07A, A10A, A12B, N02A, **J01M**, C02A, B01A, A03B, N05C, C09A, D01A, B02B, N05B, R05C, A03F, R01A, D11A, C01E, A07D (TP=17, FN=9, FP=8) |
| | DrugRec | N02B, A01A, A06A, B05C, A12C, C01C, A07A, A10A, C07A, C03C, A12B, N02A, N06A, B01A, A03B, C10A, C01B, N05C, C09A, C01D, R03A, N05B, R01A (TP=21, FN=5, FP=2) |
| 3rd Visit
Diag.: 4589, 311, 2724, 41071, 4280, 41401, 25000, 4439, 4019, 5180, 2639, 2851, E8781, 27541, 51881, 9971, E8497, 42821, V103, 3962, 78052
Prod: 0066, 3606, 3723, 8856, 9671, 0045, 0040
Sypt.: cough, depression, chest pain, vomiting, fever, nausea, bleeding | Ground Truth | N02B, A01A, A02B, A06A, B05C, C01C, A07A, C07A, C03C, A12B, C02D, N06A, B01A, C10A, N05C, C09A |
| | COGNet | N02B, A01A, A02B, B05C, A12C, A07A, A10A, N02A, N06A, A02A, **J01M**, A03B, C10A, C01B, C09A, C01D, R03A, N05B (TP=8, FN=8, FP=10) |
| | DrugRec (w/o Late) | N02B, A01A, A02B, A06A, A12A, A07A, N01A, A12B, C02A, A11C, N05C, C09A, C01D, B03B, D07A, N05B, R01A (TP=8, FN=8, FP=9) |
| | DrugRec | N02B, A01A, A02B, A06A, B05C, A07A, N01A, A12B, C02D, B01A, C01B, N05C, C09A, R03A, N05B (TP=11, FN=5, FP=4) |

## D.3 Simulation Study

We also study the performance of DrugRec on simulated data. We create a random synthetic dataset for simulation studies. The generation process of simulated data is detailed in Algorithm 2. Specifically, we simulated 5000 pseudo patients and divided them into training, validation and test set with a ratio of 4:1:1. According to the statistics of MIMIC-III, the space sizes of the symptoms, diagnoses, procedures and medications are set to 428, 1895, 1378 and 112, respectively. The hidden size $H$ is 64 and all controlling weights are set to 0.5.

**Algorithm 2** Generating Process for Simulated data

---

**Input**: Number of pseudo patients: $N_{sim}$, number of historical visits: $k$, space sizes of symptoms, diagnoses, procedures, medications: $|S|, |D|, |P|, |M|$. hidden size: $H$, controlling weights: $w_{cs}$, $w_{sd}, w_{sp}, w_{cy}$.
**Output**: The simulated data $\mathbf{X}$.

 1: Draw the drug representations: $M = \text{rand}(|M|, H)$
 2: Draw the weight matrices: $W_{cs} = \text{rand}(H, |S|)$, $W_{sd} = \text{rand}(|S|, |D|)$, $W_{sp} = \text{rand}(|S|, |P|)$
 3: Draw the weight matrices: $W_{sr} = \text{rand}(|S|, H)$, $W_{dr} = \text{rand}(|D|, H)$, $W_{pr} = \text{rand}(|P|, H)$
 4: **for** $j = 1$ to $N_{sim}$ **do**
 5:     Draw the number of visits: $T_j \sim \text{Poisson}(0.5) + 2$
 6:     Draw the confounder: $c^{(j)} = \text{rand}(T_j, H)$
 7:     Draw the initial symptom representations: $s^{(j)} = w_{cs} \cdot c^{(j)} W_{cs} + (1 - w_{cs}) \cdot \text{rand}(T_j, |S|)$
 8:     Draw the initial diagnosis representations: $d^{(j)} = w_{sd} \cdot s^{(j)} W_{sd} + (1 - w_{sd}) \cdot \text{rand}(T_j, |D|)$
 9:     Draw the initial procedure representations: $p^{(j)} = w_{sp} \cdot s^{(j)} W_{sp} + (1 - w_{sp}) \cdot \text{rand}(T_j, |P|)$
 10:     Calculate the patient visit representations: $r^{(j)} = \text{mean}(s^{(j)} W_{sr}, \ d^{(j)} W_{dr}, \ p^{(j)} W_{pr})$
 11:     Calculate the initial scores: $q^{(j)} = (w_{cy} \cdot c^{(j)} + (1 - w_{cy}) \cdot r^{(j)}) \cdot M^T$
 12:     **for** $t = 1$ to $T_j$ **do**
 13:         Update the symptom representation: $\widetilde{s}_t^{(j)} = \text{mean}(s_t^{(j)}, s_{t-1}^{(j)}, \cdots, s_{t-k}^{(j)})$
 14:         Draw the number of symptoms: $n_s = \text{randint}(2, 15)$
 15:         Sort $\widetilde{s}_t^{(j)}$ and obtain the top $n_s$ symptom indices $\widetilde{s}_t^{(j)}[: n_s]$
 16:         Update the diagnosis representation: $\widetilde{d}_t^{(j)} = \text{mean}(d_t^{(j)}, d_{t-1}^{(j)}, \cdots, d_{t-k}^{(j)})$
 17:         Draw the number of diagnoses: $n_d = \text{randint}(5, 15)$
 18:         Sort $\widetilde{d}_t^{(j)}$ and obtain the top $n_d$ diagnosis indices $\widetilde{d}_t^{(j)}[: n_d]$
 19:         Update the procedure representation: $\widetilde{p}_t^{(j)} = \text{mean}(p_t^{(j)}, p_{t-1}^{(j)}, \cdots, p_{t-k}^{(j)})$
 20:         Draw the number of procedures: $n_p = \text{randint}(2, 10)$
 21:         Sort $\widetilde{p}_t^{(j)}$ and obtain the top $n_p$ procedure indices $\widetilde{p}_t^{(j)}[: n_p]$
 22:         Update the scores: $\widetilde{q}_t^{(j)} = \text{mean}(q_t^{(j)}, q_{t-1}^{(j)}, \cdots, q_{t-k}^{(j)})$
 23:         Draw the number of medications: $n_y = \text{randint}(10, 20)$
 24:         Sort $\widetilde{q}_t^{(j)}$ and obtain the top $n_y$ medication indices $\widetilde{q}_t^{(j)}[: n_y]$
 25:     **end for**
 26:     $\mathbf{X}^{(j)} = \{\widetilde{s}^{(j)}[: n_s], \ \widetilde{d}^{(j)}[: n_d], \ \widetilde{p}^{(j)}[: n_p], \ \widetilde{q}^{(j)}[: n_y]\}$
 27: **end for**
 28: **return $\mathbf{X}$**

---

The key to the simulation is to conditionally sample the variables in sequence based on our proposed causal graph (Figure 1(b)). We compare DrugRec with COGNet and our ablation model DrugRec (w/o $\mathcal{L}_{ate}$) ignoring the hidden confounder. All methods are trained and tested on the same split of simulated data. The results of the effectiveness metrics on simulated data are shown in Table 6. DrugRec can achieve better performance than those ignoring the hidden confounder, indicating the impact of modeling the hidden confounder.

Table 6: Effectiveness results on simulated data.

| Method | Jaccard | PRAUC | F1 |
|---|---|---|---|
| COGNet | 0.8211 | 0.9598 | 0.8959 |
| DrugRec (w/o Late) | 0.8334 | 0.9680 | 0.9047 |
| DrugRec | **0.8401** | **0.9779** | **0.9091** |

### D.4 Error Analysis

We sort the prediction results of all cases by F1 score, and select the case with the lowest F1 score for analysis, which is shown in Table 7. In this case, the model recommends a total of 13 drugs, and there are 8 drugs in ground truth. The intersection of the two is only 3 drugs. We note that there are

only 3 diagnosis codes and one procedure code for the patient in this case, significantly lower than the average of test set (10.24 and 3.85). Thus, we analyze that the poor model recommendation effect may be due to insufficient observed information for inference. But since we take unseen observations into account in the confounder, the precision of our model is still stronger than the baseline COGNet. Lower false positives (FP) indicate that DrugRec is less prone to recommending too many wrong drugs.

Table 7: A bad case of drug recommendations.

| Patient Disease Condition | Method | Recommended Drugs (ATC3) |
|---|---|---|
| Diag.: 2851, 56881, V6441 | Ground Truth | A01A, N02A, A02A, J01M, B01A, N05B, C01E, D04A |
| Prod.: 0331 | COGNet | N02B, **A01A**, A02B, A06A, B05C, A12A, A12C, A07A, N01A, C07A, A12B, **N02A**, N06A, N05C, J01D, N03A, N05A, A04A, **N05B** (TP=3, FN=5, FP=16) |
| | DrugRec | N02B, **A01A**, A02B, A06A, B05C, A12C, A12B, N06A, **B01A**, N05C, J01D, A04A, **N05B** (TP=3, FN=5, FP=10) |