# OpenReview forum: "Debiased, Longitudinal and Coordinated Drug Recommendation through Multi-Visit Clinic Records"
_NeurIPS.cc/2022/Conference — NeurIPS 2022 Accept_

### Official Review · Reviewer_qNaz · 2022-07-19

**Rating:** 5
**Confidence:** 3
**Soundness:** 3 good
**Presentation:** 2 fair
**Contribution:** 3 good

**Summary:**

This paper proposes a causal inference-based drug recommendation model by considering three key factors in EHRs. The proposed two modeling schemes (DrugRec-a and DrugRec-k) handle the multi-visit patients to better model a patient’s historical health condition. And controllable DDI with 2-SAT is proposed to coordinate the recommendation and improve the recommendation safety. It is well-organized. The extensive experiments demonstrate the proposed model's effectiveness.

**Questions:**

1. How do the experiments conduct on MIMIC-IV-related symptoms?
2. How do previous methods consider three essential factors in a drug recommendation model?

**Ethics Review Area:**

["Privacy and Security (e.g., consent)"]

**Limitations:**

1. It is claimed that the symptoms are extracted from the clinical notes. As the clinical notes are generated at the discharge stage, it means medications have been given to patients at the current visit. So it is confusing to use the symptoms as inputs at the current visit.
2. It would be better to provide more information about how previous research addressed the limitations without the three key factors.
3. It seems MIMIC-IV does not have clinical notes. How do the experiments conduct on MIMIC-IV-related symptoms?
4. The proposed model is complicated, it would be better to give a complexity analysis.

**Strengths And Weaknesses:**

Strengths:
1. The paper proposes a causal inference-based model to address the drug recommendation task and provides causal analysis.
2. DrugRec-a and DrugRec-k handle the multi-visit patients to better model a patient’s historical health condition.
3. Extensive experiments on real-world datasets MIMIC-III and MIMIC-IV evaluate the recommendation quality and the DDI rate.

Weaknesses:
1. It is claimed that the symptoms are extracted from the clinical notes. As the clinical notes are generated at the discharge stage, it means medications have been given to patients at the current visit. So it is confusing to use the symptoms as inputs at the current visit.
2. It would be better to provide more information about how previous research addressed the limitations without the three key factors.
3. It seems MIMIC-IV does not have clinical notes. How do the experiments conduct on MIMIC-IV-related symptoms?

---

> ### Author Response · Authors · 2022-08-02
> **Response to Reviewer qNaz**
>
> We sincerely thank you for your constructive suggestions and valuable comments! We will answer the questions in details and will appreciate very much if you could kindly raise your score if your concerns are addressed.
>
> *Q1: "...confusing to use the symptoms as inputs at the current visit."*
>
> A1: From the current definition of the drug recommendation task, it is assumed that the information (i.e., symptoms, drugs, etc.) of the patient in a specific visit can be obtained at the same time. We also follow such an assumption of previous work [5] to leverage the patient's current symptom information for drug recommendation.
>
> *Q2: "how previous research addressed the limitations without the three key factors."*
>
> A2: We have discussed the previous studies in the section of Related Work, and this section is updated based on your great advice!
>
> i) multi-visits: previous works basically utilize RNN or memory networks to model the information in historical records [1, 2, 3]. Here we use causal graphical models for the multi-visit scenario.
>
> ii) DDI: previous works leverage the knowledge in the DDI graph and add DDI loss in the training objective [3, 4]. We model the DDI problem as the 2-satisfiability (SAT) problem, and optimize the 2-SAT based on predicted recommendation probabilities.
>
> iii) bias: to the best of our knowledge, we are the first to formulate bias as a confounder through causal effect in drug recommendation.
>
> *Q3: "How do the experiments conduct on MIMIC-IV-related symptoms?"*
>
> A3: MIMIC-IV has not officially released notes. In practice, we extracted the symptoms from the database of MIMIC-IV-ED [6] and aligned both datasets through the linkage such as `subject_id` between MIMIC-IV and MIMIC-IV-ED. With our efforts of data alignment, the enriched datasets will be released to the community to facilitate further studies.
>
> *Q4: "complexity analysis"*
>
> A4: We add a complexity analysis of previous methods and our proposed DrugRec. All experiments are implemented under the same device environment (two Nvidia A40 GPUs). The comparison results of complexity analysis are shown in the following table. Although our method is slightly slower than previous methods in training time, the efficiency in the inference stage is to some extent on a par with baselines.
>
> Although efficiency in training is an important factor to otpimize as our future work, currently we still put our priority in effectiveness improvement with acceptable inference time.
> |Model|# of Param.|Training Time (s)|Inference Time (s)|
> |:-|:-:|:-:|:-:|
> |LEAP|171,708|443.09|42.65|
> |RETAIN|444,346|100.58|17.09|
> |GAMENet|604,218|198.26|34.09|
> |MICRON|263,152|155.27|22.45|
> |SafeDrug|534,198|361.35|33.48|
> |COGNet|1,328,531|485.36|170.04|
> |DrugRec|2,127,168|567.68|41.03|
>
> [1] Choi, Edward, et al. "Retain: An interpretable predictive model for healthcare using reverse time attention mechanism."
>
> [2] Shang, Junyuan, et al. "Gamenet: Graph augmented memory networks for recommending medication combination."
>
> [3] Yang, Chaoqi, et al. "Change Matters: Medication Change Prediction with Recurrent Residual Networks."
>
> [4] Yang, Chaoqi, et al. "Safedrug: Dual molecular graph encoders for safe drug recommendations."
>
> [5] Tan, Yanchao, et al. "4SDrug: Symptom-based Set-to-set Small and Safe Drug Recommendation."
>
> [6] https://physionet.org/content/mimic-iv-ed/2.0/

---

### Official Review · Reviewer_8K1k · 2022-07-19

**Rating:** 7
**Confidence:** 4
**Soundness:** 3 good
**Presentation:** 3 good
**Contribution:** 3 good

**Summary:**

This is an important area of research for clinical/medical innovation. The authors propose DrugRec, a causal inference-based drug recommendation model. The causal graphical model can identify and deconfound the recommendation bias with front-door adjustment. In addition, the authors model the drug-drug interactions (DDIs) as the propositional satisfiability (SAT) problem and solving the SAT problem can help better coordinate the recommendation. Experiments are performed using real-world clinical data.


**Questions:**

some important assumptions are not discussed and are worth mentioning and further expanding upon:
1.	A key point that is not discussed in the assumptions is the time intervals between the visits, especially when historical data are used. For instance, having 5 visits over the past 5 years is very different from having 5 visits during the past month.
2.	Frequent refills (indication of safety and chronic conditions), and dosage adjustment are important elements in the recommendations that are also not discussed.
3.	Medication corrections – a modification that is only a few days apart can easily be corrected and provide better quality data in the pre-processing.
4.	Medication recommendations for in-patients vs. out-patients are very different; which one is being modeled here? The modeling assumption and data pre-processing need a different level of attention depending on the target population.
5.	Regarding the use of ICD codes. Often the ICD codes are used with a rule of 2, especially for complex conditions that require further investigation to confirm the correct diagnosis.
6.	Natural language processing: it is not clear how symptoms were extracted from clinical notes. Also, there are different types of notes, and it is not clear what type of notes were used at what time during the encounter.

**Limitations:**

A number of assumptions and clarifications are lacking, limiting the scope and utility of the study.

**Strengths And Weaknesses:**

Strengths: Use of real-world data, even though MIMIC has certain characteristics and does not represent typical EHR data for general patients.
Weaknesses: A number of assumptions are not discussed or addressed. See sections "Questions"

---

> ### Author Response · Authors · 2022-08-02
> **Response to Reviewer 8K1k**
>
> We sincerely thank you for your constructive suggestions and valuable comments! We will answer the questions in details and will appreciate very much if you could kindly support our work during the discussion phase.
>
> *Q1: "time intervals between the visits"*
>
> A1: Thanks for your constructive comments! Generally, we follow the traditional drug recommendation paradigm based on the assumption that treats multiple visits of a patient as the same time interval [1, 2, 3]. Your suggestion makes so much sense to us, and we will definitely try the new recommendation paradigm considering the time period as our future work!
>
> *Q2: "Frequent refills (indication of safety and chronic conditions), and dosage adjustment are important elements in the recommendations that are also not discussed."*
>
> A2: Thanks for your valuable comments! These mentioned factors are indeed interesting and of great importance. Since the data lack of these elements, we list the investigation of refills and dosage adjustment as our future work.
>
> *Q3: "Medication corrections provide better quality data in the pre-processing."*
>
> A3: Thanks for your valuable comments! We believe that medication corrections can have a great impact on drug recommendations. We will consider medication corrections when data are valid in our future work.
>
> *Q4: "Medication recommendations for in-patients vs. out-patients?"*
>
> A4: The two MIMIC datasets we use are both based on in-patients [7, 8], which provided critical care data admitted to intensive care units.
> Therefore, the modeling assumption and data pre-processing in our work are consistent.
>
> *Q5: "the ICD codes are used with a rule of 2, especially for complex conditions"*
>
> A5: Thanks for your advice to improve our work! The data we use is anonymized patient information. We will verify the reliability of the ICD codes with your mentioned rule when data are available.
>
> *Q6: "how symptoms were extracted from clinical notes? what type of notes were used at what time during the encounter"*
>
> A6: We will definitely add more elaborations in the revision.
>
> i) Following previous works on the applications of clinical notes [4, 5, 6], we filtered the clinical notes with the category of "Discharge summary". It is considered the best single note type for extracting biomedical concepts like symptoms.
>
> ii) We collected as many canonical symptom terms as possible and constructed symptom sets. Next, following the work of [6], we extract symptoms from clinical notes using template based partial matching, which is of practical use in this work.
>
> [1] Choi, Edward, et al. "Retain: An interpretable predictive model for healthcare using reverse time attention mechanism."
>
> [2] Shang, Junyuan, et al. "Gamenet: Graph augmented memory networks for recommending medication combination."
>
> [3] Yang, Chaoqi, et al. "Safedrug: Dual molecular graph encoders for safe drug recommendations."
>
> [4] Long, William. "Extracting diagnoses from discharge summaries."
>
> [5] Steinkamp, Jackson M., et al. "Task definition, annotated dataset, and supervised natural language processing models for symptom extraction from unstructured clinical notes."
>
> [6] Zeng, Xian, et al. "PIC, a paediatric-specific intensive care database."
>
> [7] https://physionet.org/content/mimiciii/1.4/
>
> [8] https://physionet.org/content/mimiciv/2.0/

---

### Official Review · Reviewer_ecUC · 2022-07-24

**Rating:** 6
**Confidence:** 2
**Soundness:** 2 fair
**Presentation:** 2 fair
**Contribution:** 2 fair

**Summary:**

The paper address the task of drug recommendation from EHR data. The authors propose a causal inference graphical model with front-door adjustment to address the issue of hidden confounders, which leads to bias. The proposed model handles data with multiple visits (DrugRec-a: averaging visits, DrugRec-k: taking the last k visits) by extending the causal graph to link consecutive visits. The authors propose a method to tackle drug-drug interaction (DDI) in the recommender system by solving a propositional satisfiability (SAT) problem. Experiments on MIMIC datasets show the effectiveness of the model based on various metrics.

**Questions:**

What is the distribution of number of visits in the datasets?
The role of k could be better highlighted by stratifying the datasets based on different number of visits.

Typos:
L299: addtion -> addition
L319: methrics -> metrics



**Limitations:**

The experiments are based on two MIMIC datasets which are likely overlapping in data


**Strengths And Weaknesses:**

Strengths:
- The paper addresses both hidden confounders and DDI in a single model
- The paper includes a causal modeling of the sequential aspect of EHR
Weakness:
- The novelty is rather incremental and a combination of known methods
-  A simulation example could be added to better assess the hidden confounders

---

> ### Author Response · Authors · 2022-08-02
> **Response to Reviewer ecUC**
>
> We sincerely thank you for your constructive suggestions and valuable comments! We will answer the questions in details and will appreciate very much if you could kindly raise your score if your concerns are addressed.
>
> *Q1: “novelty”*
>
> A1: Our contributions are three-fold in terms of novelty:
>
> i) To the best of our knowledge, we are the first to model the drug recommendation problem using causal formulation with bias alleviation. It is a new perspective to tackle the problem, with prominent performance improvement achieved (Appendix Tables 2-3, lines 76-79).
>
> ii) Our proposed approach is also featured with longitudinal modeling of casual recommendation based on previous patient visits and historical health condition. It is a real-world scenario and we solve the longitudinal graphical model with multi-visits through new inference methods, which is another contribution.
>
> iii) In addition, we model the drug-drug interactions as a 2-satisfiability problem to control the safety of drug combinations. It is non-trivial to solve the 2-SAT optimization.
>
> *Q2: “A simulation example”.*
>
> A2: i) Thanks for your suggestions! We created a random synthetic dataset for simulation studies. The generation process of simulated data is detailed in Algorithm 2 in Appendix D.3 (L111). The key to the simulation is to conditionally sample the variables in sequence based on our proposed causal graph (Figure 1). We compare DrugRec with COGNet and our ablation model DrugRec (w/o $\mathcal{L}_{ate}$)  ignoring the hidden confounder. All methods are trained and tested on the same split of simulated data. The results of the effectiveness metrics on simulated data are shown in the following table. DrugRec can achieve better performance than those ignoring the hidden confounder, indicating  the impact of modeling the hidden confounder.
>
> |Method|Jaccard|PRAUC|F1|
> |:-|:-:|:-:|:-:|
> |COGNet|0.8211|0.9598|0.8959|
> |DrugRec (w/o $\mathcal{L}_{ate}$)|0.8334|0.9680|0.9047|
> |DrugRec|0.8401|0.9779|0.9091|
>
> ii) To intuitively validate the impact of modeling the hidden confounder, we have also provided a drug recommendation example for a patient with three visits in Table 5 in Appendix D.2 (lines 89-104). In later visits (2nd, 3rd), the baselines may accumulate recommendation bias. In this case, COGNet and DrugRec (w/o $\mathcal{L}_{ate}$) always predict a drug "J01M" that is frequently recommended in the training set, but is not included in the current ground truth. This makes drug recommendations more biased. Since DrugRec models the hidden confounder, it will not always recommend the wrong drug and achieve debiased drug recommendations, improving the recommendation accuracy.
>
> *Q3: "...the distribution of number of visits...? The role of k by stratifying the datasets based on different number of visits."*
>
> A3: i) We have included the histograms of number of visits on MIMIC-III and MIMIC-IV in the revision of Appendix C. The average number of visits for different patients in MIMIC-III and MIMIC-IV are 2.59 and 2.90, respectively. The proportion of patients with more than 3 visits in two datasets do not exceed 10%.
>
> ii) Thanks for your suggestions! We stratified the datasets based on different number of visits to study its impact on the performance of different models. Following the preprocessing scripts of [1], patients with only 1 visit were removed. Thus, we stratified the test set into 3 groups by the total number of patient visits: 2, 3 and more than 3 visits. Here we chose the recent COGNet and SafeDrug as stronger baselines. The comparison results of various methods on different number of visits are as follows. Our DrugRec outperformed baselines on all metrics in each group. For different values of $k$ in DrugRec, DrugRec-2 has the best performance, which concurs with the conclusion in Section Experiments.
>
> We will add more details of this study in the revision.
>
> |Method||Jaccard|||PRAUC|||F1||
> |:-|:-:|:-:|:-:|:-:|:-:|:-:|:-:|:-:|:-:|
> ||2|3|>3|2|3|>3|2|3|>3|
> |SafeDrug|0.5114|0.5016|0.5006|0.7653|0.7652|0.7587|0.6678|0.6620|0.6617|
> |COGNet|0.5131|0.5146|0.5140|0.7675|0.7689|0.7674|0.6703|0.6722|0.6705|
> |DrugRec-1|0.5216|0.5170|0.5146|0.7687|0.7681|0.7674|0.6775|0.6731|0.6724|
> |DrugRec-2|0.5237|0.5176|0.5147|0.7729|0.7691|0.7690|0.6782|0.6739|0.6729|
> |DrugRec-3|0.5225|0.5170|0.5147|0.7720|0.7683|0.7680|0.6774|0.6738|0.6732|
>
> *Q4: "Typos."*
>
> A4: Thank you for the careful proofreading! We will correct these typos in the revision.
>
> *Q5: "The experiments are based on two MIMIC datasets which are likely overlapping in data."*
>
> A5: We confirm that there is no overlap between the two MIMIC datasets. The ICD codes from the two datasets are converted into textual descriptions with different dictionary tables. We have also double-checked the contents and IDs and thus guaranteed neither the training set nor the test set in our experiments have any overlap.
>
> [1] Yang, Chaoqi, et al. "Safedrug: Dual molecular graph encoders for safe drug recommendations."

---

> > ### Comment · Reviewer_ecUC · 2022-08-08
> > **updated rating**
> >
> > Thank you for addressing my concerns. I've updated my rating accordingly.

---

### Official Review · Reviewer_wUqG · 2022-07-25

**Rating:** 6
**Confidence:** 4
**Soundness:** 3 good
**Presentation:** 4 excellent
**Contribution:** 3 good

**Summary:**

In this paper, the authors proposed a causal model, DrugRec, for drug recommendation. The model utilizes different historic and clinical information to target three specific problems: (1) elimination of drug recommendation bias (2) better utilisation of historic patient data (3) use of the clinical information to reduce harmful drug-drug interaction (DDI). There are two variants of DrugRec which are compared against multiple baseline models and they outperformed all the existing baselines over the publicly available MIMIC dataset.

**Questions:**

Copied from weaknesses:
2. Reference Figure 1: Why is M not connected to S? M^j_t is the multi-hot encoded medication set of the jth patient right? Because D and P would not cover what medications or drugs were previously given to the patients in earlier visits.

3. Ref DrugRec-k: Why did the authors choose k visits, rather than a specific time period before the visit? This provides different context windows for different patients. One patient can have a lookback period of 5 years whereas other can have of 5 months.

4. (i) Reference L249-251: If you did not want to use ICD coding, a better way would have been to use an entity normalizer to identify which
        ICD codes belong to the same set of families. ICD codes also have a hierarchy provided by UMLS or other clinical meta-thesaurus.
    (ii) Secondly, how was negation handled here? Such as in scenarios like "The pt. did not have XYZ ......"

5. The limitations of the proposed model are not discussed and there is no specific error analysis to discuss the short-comings of the model or future work. Could authors elaborate a bit as to what are the potential limitations of their proposed model?

**Limitations:**

The limitations are not discussed by the authors.
1. The Related work section can be improved by clearly identifying the difference between the proposed model and the class of related works. Each para that identifies a particular direction of work can have a line or two to identify the difference.
2. Limitation and error analysis is not discussed in the paper.
3. It would be great if authors could add the algo of their proposed model in the paper.

**Strengths And Weaknesses:**

This is a strong paper but have some (fundamental) weaknesses that I would like the authors to address during the rebuttal period.

Strengths:
1. The paper is neatly written with very few grammatical errors and is easy to follow.
2. DrugRec is compared against all the appropriate baselines for drug recommendation.
3. I appreciate that the authors have tried to justify their decisions at all steps. (I have a few questions regarding some of them below)
4. The ablation study is important and shows that not only the historic patient data is important but also the way in which they are modelled makes a significant difference in the performance.

Weakness/Questions:
1. The Related Works section can be improved a lot, the authors should discuss how their model is different from the existing work cited in the section. 1-2 liner discussions in each section of Related Works would also be enough. This helps the reader in identifying the clear advantage of the proposed model.
2. Reference Figure 1: Why is M not connected to S? M^j_t is the multi-hot encoded medication set of the jth patient right? Because D and P would not cover what medications or drugs were previously given to the patients in earlier visits.
3. Ref DrugRec-k: Why did author choose k visits, rather than a specific time period before the visit? This provides different context windows for different patients. One patient can have a lookback period of 5 years whereas other can have of 5 months.
4. (i) Reference L249-251: If you did not want to use ICD coding, a better way would have been to use an entity normalizer to identify which
        ICD codes belong to the same set of families. ICD codes also have a hierarchy provided by UMLS or other clinical meta-thesaurus.
    (ii) Secondly, how was negation handled here? Such as in scenarios like "The pt. did not have XYZ ......"
5. The limitations of the proposed model are not discussed and there is no specific error analysis to discuss the short-comings of the model or future work.

---

> ### Author Response · Authors · 2022-08-02
> **Response to Reviewer wUqG**
>
> We sincerely thank you for your constructive suggestions and valuable comments! We will answer the questions in details and will appreciate very much if you could kindly raise your score to support our work during the discussion phase.
>
> *Q1. "...improve the Related Works section..."*
>
> A1: We have included the discussion about the existing works and updated our submission according to your suggestion. Thanks for your advice!
>
> *Q2: "Why is M not connected to S?"*
>
> A2: In Figure 1, $S$ represents the patient's symptoms, $M$ represents a drug from the drug candidates, and $Y$ represents whether the drug $M$ is recommended. Based on the causal perspective, the patient's current symptoms cannot be directly causally related to all possible drug candidate of $M$, while it should be connected through a path leading to the recommended decision $Y$.
>
> *Q3: "why choose k visits rather than a specific time period".*
>
> A3: Yes it is an inspiring suggestion! Generally, we follow the traditional drug recommendation paradigm based on the assumption that treats multiple visits of a patient as the same time interval [1, 2, 3].
> Your suggestion makes so much sense to us, and we will definitely try the new recommendation paradigm considering the time period as our future work!
>
> *Q4: "(i) use an entity normalizer; (ii) how to handle negation"*
>
> A4: i) Currently, we use the text description which represents the ICD codes in terms of literal and original semantic meaning. We would agree that an entity normalizer could be another way to learn the ICD representation with hierarchies. Thank you again for this advice!
>
> ii) According to our statistics, there are less than 5% samples with negation terms modifying symptoms, which means low coverage. We will optimize the symptom extraction method to eliminate effects of negation in the future.
>
> *Q5: "The limitations of the proposed model, error analysis and future work"*
>
> A5: i) **Limitation:** since our method samples symptoms in the training step due to causal effect identification, it is slightly less efficient than previous lightweight models. Efficiency and effectiveness are generally in trade-off and we now include the efficiency analysis in the Appendix D.4 (lines 118-124).
>
> ii) **Error analysis:**  we sort the prediction results of all cases by F1 score, and select the case with the lowest F1 score for analysis, which is shown as follows. In this case, the model recommends a total of 13 drugs, and there are 8 drugs in ground truth. The intersection of the two is only 3 drugs. We note that there are only 3 diagnosis codes and one procedure code for the patient in this case, significantly lower than the average of test set (10.24 and 3.85). Thus, we analyze that the poor model recommendation effect may be due to insufficient observed information for inference. But since we take unseen observations into account in the confounder, the precision of our model is still stronger than the baseline COGNet. Lower false positives (FP) indicate that DrugRec is less prone to recommending too many wrong drugs.
>
> |Patient Disease Condition|Method|Recommended Drugs (ATC3)|
> |-|-|-|
> |Diag.: 2851, 56881, V6441|Ground Truth|A01A, N02A, A02A, J01M, B01A, N05B, C01E, D04A|
> |Prod.: 0331|COGNet|N02B, **A01A**, A02B, A06A, B05C, A12A, A12C, A07A, N01A, C07A, A12B, **N02A**, N06A, N05C, J01D, N03A, N05A, A04A, **N05B** (TP=3, FN=5, FP=16)|
> ||DrugRec|N02B, **A01A**, A02B, A06A, B05C, A12C, A12B, N06A, **B01A**, N05C, J01D, A04A, **N05B** (TP=3, FN=5, FP=10)|
>
> iii) **Future work:** we will try to model the task in the new recommendation paradigm based on time period and we will also test the entity normalizer for ICD encoding while we will continue to improve the efficiency of our proposed model.
>
> *Q6: "the algo of their proposed model"*
>
> A6: The training process for our model is shown in Algorithm 1 in Appendix B.3 (lines 29-31). After obtaining the input representations of a patient's current diagnoses, procedures and symptoms, we use updating network to derive the current patient visit representations. Then the scoring network is used to score each candidate drug and make eventual drug recommendations.
>
> [1] Choi, Edward, et al. "Retain: An interpretable predictive model for healthcare using reverse time attention mechanism."
>
> [2] Shang, Junyuan, et al. "Gamenet: Graph augmented memory networks for recommending medication combination."
>
> [3] Yang, Chaoqi, et al. "Safedrug: Dual molecular graph encoders for safe drug recommendations."

---

> > ### Comment · Reviewer_wUqG · 2022-08-07
> > **Response to Author Rebuttal**
> >
> > Thank you for looking into the concerns/suggestions and including the ones which were possible in the current timeline. I do not have any further concerns. I believe my current overall recommendation would remain the same for the paper. Thank you.

---

### Meta-Review · Area_Chair_LsDA · 2022-08-24

**Recommendation:** Accept
**Confidence:** Certain

**Metareview:**

This paper proposes a drug recommendation model, called DrugRec, based on causal inference. The two proposed modeling schemes are capable of handling multiple patient visits to better model the patient's past health status. The authors performed large-scale experiments and showed the effectiveness of the proposed model. The model formulates bias as a confounder through causal effect. The authors clearly answered the questions from the reviewers, and so the revised paper would be an achievement worthy of being accepted as a new NeurIPS paper.

**Award:**

No

---

### Decision · Program_Chairs · 2022-09-14

Accept